# Precise characterization of the prior predictive distribution of deep ReLU networks

**Lorenzo Noci**[*]
Dept of Computer Science
ETH Zürich
lorenzo.noci@inf.ethz.ch

**Gregor Bachmann**[*]
Dept of Computer Science
ETH Zürich
gregor.bachmann@inf.ethz.ch

**Kevin Roth**[*]
Dept of Computer Science
ETH Zürich
kevin.roth@inf.ethz.ch

**Sebastian Nowozin**
Microsoft Research
Cambridge, UK
Sebastian.Nowozin@microsoft.com

**Thomas Hofmann**
Dept of Computer Science
ETH Zürich
thomas.hofmann@inf.ethz.ch

## Abstract

Recent works on Bayesian neural networks (BNNs) have highlighted the need to better understand the implications of using Gaussian priors in combination with the compositional structure of the network architecture. Similar in spirit to the kind of analysis that has been developed to devise better initialization schemes for neural networks (cf. He- or Xavier initialization), we derive a precise characterization of the prior predictive distribution of finite-width ReLU networks with Gaussian weights. While theoretical results have been obtained for their heavy-tailedness, the full characterization of the prior predictive distribution (i.e. its density, CDF and moments), remained unknown prior to this work. Our analysis, based on the Meijer-G function, allows us to quantify the influence of architectural choices such as the width or depth of the network on the resulting shape of the prior predictive distribution. We also formally connect our results to previous work in the infinite width setting, demonstrating that the moments of the distribution converge to those of a normal log-normal mixture in the infinite depth limit. Finally, our results provide valuable guidance on prior design: for instance, controlling the predictive variance with depth- and width-informed priors on the weights of the network.

## 1 Introduction

It is well known that standard neural networks initialized with Gaussian weights tend to Gaussian processes (Rasmussen, 2003) in the infinite width limit (Neal, 1996; Lee et al., 2018; de G. Matthews et al., 2018), coined *neural network Gaussian process* (NNGP) in the literature. Although the NNGP has been derived for a number of architectures, such as convolutional (Novak et al., 2019; Garriga-Alonso et al., 2019), recurrent (Yang, 2019) and attention mechanisms (Hron et al., 2020), little is known about the finite width case.

The reason why the infinite width limit is relatively tractable to study is that uncorrelated but dependent units of intermediate layers become normally distributed due to the central limit theorem (CLT) and as a result, independent. In the finite width case, zero correlation does not imply independence, rendering the analysis far more involved as we will outline in this paper.

One of the main motivations for our work is to better understand the implications of using Gaussian priors in combination with the compositional structure of the network architecture. As argued by Wilson and Izmailov (2020); Wilson (2020), the prior over parameters does not carry a meaningful

35th Conference on Neural Information Processing Systems (NeurIPS 2021).

interpretation; the prior that ultimately matters is the *prior predictive distribution* that is induced when a prior over parameters is combined with a neural architecture (Wilson and Izmailov, 2020; Wilson, 2020).

Studying the properties of this prior predictive distribution is not an easy task, the main reason being the *compositional* structure of a neural network, which ultimately boils down to products of random matrices with a given (non-linear) activation function. The main tools to study such products are the Mellin transform and the Meijer-G function (Meijer, 1936; Springer and Thompson, 1970; Mathai, 1993; Stojanac et al., 2017), both of which will be leveraged in this work to gain theoretical insights into the inner workings of BNN priors.

**Contributions** Our results provide an important step towards understanding the interplay between architectural choices and the distributional properties of the prior predictive distribution, in particular:

- We characterize the prior predictive density of finite-width ReLU networks of any depth through the framework of Meijer-G functions.
- We draw analytic insights about the shape of the distribution by studying its moments and the resulting heavy-tailedness. We disentangle the roles of width and depth, demonstrating how deeper networks become more and more heavy-tailed, while wider networks induce more Gaussian-like distributions.
- We connect our work to the infinite width setting by recovering and extending prior results (Lee et al., 2018; Matthews et al., 2018) to the infinite depth limit. We describe the resulting distribution in terms of its moments and match it to a normal log-normal mixture (Yang, 2008), empirically providing an excellent fit even in the non-asymptotic regime.
- Finally, we introduce generalized He priors, where a desired variance can be directly specified in the function space. This allows the practitioner to make an interpretable choice for the variance, instead of implicitly tuning it through the specification of each layer variance.

The rest of the paper is organized as follows: in Section 3.1, we introduce the relevant notation for the neural network that will be analyzed. We describe the prior works of Lee et al. (2018); Matthews et al. (2018) in more detail in Section 3.2 . Then, in Section 3.3, we introduce the Meijer-G function and the necessary mathematical tools. In Section 4 we derive the probability density function for a linear network of any depth and extend these results to ReLU networks, which represents the key contribution of our work. In Section 5, we present several consequences of our analysis, including an extension of the infinite width setting to infinite depth as well as precise characterizations of the heavy-tailedness in the finite regime. Finally, in Section 6, we show how one can design an architecture and a prior over the weights to achieve a desired prior predictive variance.

## 2 Related Work

Although Bayesian inference in deep learning has recently risen in popularity, surprisingly little work has been devoted to investigating standard priors and their implied implicit biases in neural architectures. Only through the lens of infinite width, progress has been made (Neal, 1996), establishing a Gaussian process behaviour at the output of the network. More recently, Lee et al. (2018) and Matthews et al. (2018) extended this result to arbitrary depth. We give a brief introduction in Section 3.2. Due to their appealing Gaussian process formulation, infinite width networks have been extensively studied theoretically, leading to novel insights into the training dynamics under gradient descent (Jacot et al., 2018) and generalization (Arora et al., 2019b).

Although theoretically very attractive, the usage of infinite width has been severely limited by its inferior empirical performance (Arora et al., 2019a). While first insights into this gap have been obtained (Aitchison, 2020; Aitchison et al., 2021), the picture is far from complete and a better understanding of finite width networks is still highly relevant for practical applications. In the finite regime, however, such precise characterizations in function space have been elusive so far and largely limited to empirical insights (Flam-Shepherd et al., 2017, 2018) and investigations of the heavy-tailedness of layers (Vladimirova et al., 2019; Fortuin et al., 2021). The field of finite-width corrections has recently gained a lot of attraction. Hanin and Nica (2020) studies the simultaneous limit of width and depth of the Jacobian of a ReLU net. More recently, a number of concurrent works

appearing shortly before or after ours, such as Zavatone-Veth and Pehlevan (2021); Roberts et al. (2021); Li et al. (2021) study properties of large-but-finite neural nets. Notably, Zavatone-Veth and Pehlevan (2021) concurrently derived similar results on the characterization of the prior predictive distribution of finite width networks, however, we did not discover them until after we had completed this work. Note also that our novel limiting behaviours (cf. Sec. 5.1) and our analytical insights into heavy-tailedness (cf. Sec. 5.2) clearly distinguish our work from theirs. In addition, our results also offer valuable guidance on prior design for ML practitioners (cf. Sec. 6). Li et al. (2021) derive a similar limiting result in the infinite-width and depth setting in the case of Resnets (ours if for fully-connected, cf. Sec. 5.1), albeit with a completely different proof technique. Moreover, we also give precise insights into the prior predictive distribution for finite width (cf. Sec. 4), whereas Li et al. (2021) only work with the limits.

Finally, this work is also related to the studies of signal propagation into finite-width random networks (Poole et al., 2016; Schoenholz et al., 2017) and initialization (He et al., 2015; Hanin and Rolnick, 2018). In particular, He et al. (2015) uses a second moment analysis to specify the variance of the weights. In this sense, our approach extends it by deriving all the moments of the distribution.

## 3 Background

### 3.1 Fully Connected Neural Network

Given an input $\boldsymbol{x} \in \mathbb{R}^d$, we define a $\boldsymbol{\theta}$-parameterized $L$ layer fully-connected neural network $f_{\boldsymbol{\theta}}(\boldsymbol{x})$ as the composition of layer-wise affine transformations $\boldsymbol{W}^{(l)} \in \mathbb{R}^{d_{l-1} \times d_l}$ and element-wise non-linearities $\sigma : \mathbb{R} \to \mathbb{R}$,

$$f_{\boldsymbol{\theta}}(\boldsymbol{x}) = \boldsymbol{W}^{(L)\top} \sigma\left(\boldsymbol{W}^{(L-1)\top} \dots \sigma\left(\boldsymbol{W}^{(1)\top} \boldsymbol{x}\right) \dots\right), \tag{1}$$

where $\boldsymbol{\theta} = (\boldsymbol{W}^{(1)}, \dots, \boldsymbol{W}^{(L)})$ denotes the collection of all weights. Throughout this work, we assume standard initialization, i.e. $W_{ij}^{(l)} \sim \mathcal{N}(0, \sigma_l^2)$, where weights in each layer can have a different variance $\sigma_l^2$. Often, it will be more convenient to work with the corresponding unit-level formulation, expressed through the recursive equations:

$$f_k^{(l)}(\boldsymbol{x}) = \sum_{j=1}^{d_{l-1}} W_{jk}^{(l)} g_j^{(l-1)}(\boldsymbol{x}) \ , \quad \boldsymbol{g}^{(l)}(\boldsymbol{x}) = \sigma(\boldsymbol{f}^{(l)}(\boldsymbol{x})), \tag{2}$$

When propagating an input $\boldsymbol{x}$ through the network we refer to $\boldsymbol{f}^{(l)}(\boldsymbol{x}) \in \mathbb{R}^{d_l}$ as the *pre-activations* and to $\boldsymbol{g}^{(l)}(\boldsymbol{x}) \in \mathbb{R}^{d_l}$ as the *post-activations*. When it is clear from the context, to enhance readability, we will occasionally abuse notation denoting $\boldsymbol{f}^{(l)} := \boldsymbol{f}^{(l)}(\boldsymbol{x})$ and $\boldsymbol{g}^{(l)} := \boldsymbol{g}^{(l)}(\boldsymbol{x})$. We will also use the short-hand $[m] = \{1, \dots, m\}$. Imposing a probability distribution on the weights induces a distribution on the pre-activations $\boldsymbol{f}^{(l)}(\boldsymbol{x})$. Understanding the properties of this distribution is the main goal of this work.

### 3.2 Prior Predictive Distribution and Infinite Width

A precise characterization of the prior predictive distribution of a neural network has been established in the so-called infinite width setting (Neal, 1995; Lee et al., 2018; Matthews et al., 2018). By considering variances that scale inversely with the width, i.e. $\sigma_l^2 = \frac{1}{d_l}$ and fixed depth $L \in \mathbb{N}$, it can be shown that the implied prior predictive distribution converges in law to a Gaussian process:

$$f_{\boldsymbol{\theta}} \xrightarrow{d} \mathcal{GP}(0, \Sigma^{(L)}), \tag{3}$$

where $\Sigma^{(L)} : \mathbb{R}^d \times \mathbb{R}^d, \ \boldsymbol{x}, \boldsymbol{x}' \mapsto \Sigma^{(L)}(\boldsymbol{x}, \boldsymbol{x}')$ is the NNGP kernel (Lee et al., 2018), available in closed-form through the recursion

$$\Sigma^{(1)}(\boldsymbol{x}, \boldsymbol{x}') = \boldsymbol{x}^T \boldsymbol{x}' \ , \quad \Sigma^{(l+1)}(\boldsymbol{x}, \boldsymbol{x}') = \mathbb{E}_{\boldsymbol{z} \sim \mathcal{N}(\mathbf{0}, \tilde{\boldsymbol{\Sigma}}^{(l)})}\left[\sigma(z_1)\sigma(z_2)\right], \tag{4}$$

for $l = 1, \dots, L-1$ and where $\tilde{\boldsymbol{\Sigma}}^{(l)} = \begin{pmatrix} \Sigma^{(l)}(\boldsymbol{x}, \boldsymbol{x}) & \Sigma^{(l)}(\boldsymbol{x}, \boldsymbol{x}') \\ \Sigma^{(l)}(\boldsymbol{x}', \boldsymbol{x}) & \Sigma^{(l)}(\boldsymbol{x}', \boldsymbol{x}') \end{pmatrix} \in \mathbb{R}^{2 \times 2}$. The proof relies on the multivariate central limit theorem in conjunction with an inductive technique, letting hidden

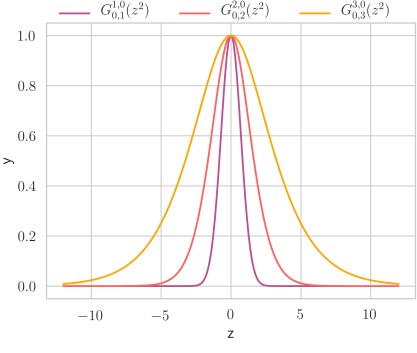
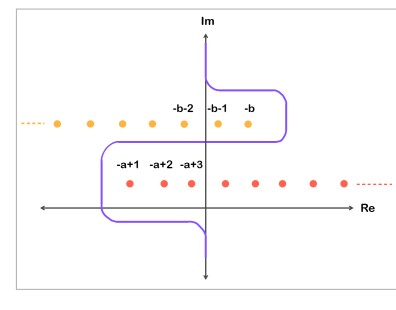

(a) Plots of Meijer-G functions  (b) Visualization of integration path

Figure 1: (a) Plot of the Meijer-G functions $G_{0,l}^{l,0}(\cdot|\boldsymbol{b})$ for $l = 1, 2, 3$ and $\boldsymbol{b} = 0, (0, 5), (0, 5, 5)$. (b) An example path $\mathcal{L}$ in the complex plane. Notice how the orange singularities are always to the left of $\mathcal{L}$ and the red ones always to the right.

layer widths go to infinity in sequence. Due to the Gaussian nature of the limit, exact Bayesian inference becomes tractable and techniques from the Gaussian process literature can be readily applied (Rasmussen, 2003). Although theoretically very appealing, from a practical point of view, infinite width networks are not as relevant due to their inferior performance (Novak et al., 2019). Gaining a better understanding of this performance gap is hence of utmost importance.

### 3.3   Meijer-G function

The Meijer-G function is the central tool to our analysis of the predictive prior distribution in the finite width regime. The Meijer-G function is a ubiquitous tool, appearing in a variety of scientific fields ranging from mathematical physics (Pishkoo and Darus, 2013) to symbolic integration software (Adamchik and Marichev, 1990) to electrical engineering (Ansari et al., 2011). Despite its high popularity in many technical fields, there have only been a handful of works in ML leveraging this elegant and convenient theoretical framework (Alaa and van der Schaar, 2019; Crabbe et al., 2020). In the following, we will introduce the Meijer-G function along with the relevant mathematical tools to develop our theory.

For $s \in \mathbb{C}$ with real-part $\mathfrak{R}(s) > 0$, denote by $\Gamma(s)$ the Gamma function defined as

$$\Gamma(s) = \int_0^\infty x^{s-1} e^{-x} dx. \tag{5}$$

We then consider the analytic continuation of $\Gamma$, which extends it to the entire complex plane, as frequently done in complex analysis. Now fix $m, n, p, q \in \mathbb{N}$ such that $0 \le m \le q$ and $0 \le n \le p$ and consider $\boldsymbol{a} \in \mathbb{R}^p$, $\boldsymbol{b} \in \mathbb{R}^q$ such that $a_i - b_j \notin \mathbb{Z}_{>0} \ \forall i = 1, \dots, p$ and $j = 1, \dots, q$. The Meijer-G function (Meijer, 1936; Mathai, 1993; Mathai and Saxena, 2006), is defined as:

$$G_{p,q}^{m,n}\left(z \Big| \begin{smallmatrix} \boldsymbol{a} \\ \boldsymbol{b} \end{smallmatrix}\right) = \frac{1}{2\pi i} \int_{\mathcal{L}} \chi(s) z^{-s} ds, \tag{6}$$

where:

$$\chi(s) = \frac{\prod_{j=1}^m \Gamma(b_j + s) \prod_{j=1}^n \Gamma(1 - a_j - s)}{\prod_{j=m+1}^q \Gamma(1 - b_j - s) \prod_{j=n+1}^p \Gamma(a_j + s)}, \tag{7}$$

and the integration path $\mathcal{L}$ defines a suitable complex curve, described in the following. Recall that the function $\Gamma(s)$ has poles at $0, -1, -2, \dots$ all the way to $-\infty$. Hence, $\Gamma(1 - a_j - s)$ has poles $-a_j + 1, -a_j + 2, \dots$, all the way to $\infty$, and $\Gamma(b_j + s)$ has poles $-b_j, -b_j - 1, \dots$, all the way to $-\infty$. The path $\mathcal{L}$ is defined such that the poles of $\Gamma(b_j + s)$ are to the left of $\mathcal{L}$ and the ones of $\Gamma(1 - a_j - s)$ are to the right of it. The condition $a_i - b_j \notin \mathbb{Z}_{>0}$ makes sure we can find such a separation as this implies that the poles do not overlap. We illustrate an example of such a path in Figure 1b. In red we display the poles of $\Gamma(1 - a_j + s)$ and in orange the poles of $\Gamma(b_j + s)$. For interested readers we refer to Beals and Szmigielski (2013) for a more extensive introduction.

The defining property of Meijer-G functions is their closure under integration, i.e. the convolution of two Meijer-G functions is again a Meijer-G function. Combined with the fact that most elementary functions can be written as a Meijer-G function, this property becomes extremely powerful at expressing complicated integrals neatly. Our proofs leverage this result extensively by expressing the integrands encountered in the prior predictive function as Meijer-G functions.

Throughout this text, we will only encounter Meijer-G functions of a simpler signature $G_{0,l}^{l,0}(\cdot|\boldsymbol{b})$. For completeness, we show its functional form here:

$$G_{0,l}^{l,0}(z|\boldsymbol{b}) = \frac{1}{2\pi i} \int_{\mathcal{L}} z^{-s} \prod_{j=1}^{l} \Gamma(b_j + s) ds. \tag{8}$$

Small values for $m \in \mathbb{N}$ correspond to familiar functions such as the exponential $G_{0,1}^{1,0}(z|0) = e^{-z}$ and the modified Bessel function of second kind $G_{0,2}^{2,0}\left(\frac{z^2}{4}\big|[\frac{\nu}{2}, -\frac{\nu}{2}]\right) = 2K_\nu(z)$. We visualize several Meijer-G functions of this form in Figure 1a. For illustrative purposes, we normalize the functions to have a maximum value of 1.

## 4 Predictive Priors for Neural Networks

In this section we detail our theoretical results on the predictive prior distribution implied by a fully-connected neural network with Gaussian weights, both with and without ReLU non-linearities.

**Linear Networks:** First, we consider linear networks, i.e. fully-connected networks where the post-activations coincide with the pre-activations. They can be characterized as the product of Gaussian random matrices, for which the result in terms of Meijer-G functions is known (consider for instance Ipsen (2015)). To highlight the differences between the linear and the non-linear approach, we re-prove the linear case leveraging our proof technique and notation. For simplicity, we assume w.l.o.g. that the input is normalized, i.e $||\boldsymbol{x}|| = 1$. We now present the resulting distribution of the predictive prior:

**Theorem 4.1.** *Suppose $l \geq 1$, and the input has dimension $d_0$. Then, the joint marginal density of the random vector $\boldsymbol{f}^{(l)}$ (i.e. the density of the $l$-th layer pre-activations) is proportional to:*

$$p(\boldsymbol{f}^{(l)}) \propto G_{0,l}^{l,0}\left(\frac{||\boldsymbol{f}^{(l)}||^2}{2^l \sigma^2}\bigg|0, \frac{1}{2}(d_1 - d_l), \ldots, \frac{1}{2}(d_{l-1} - d_l)\right), \tag{9}$$

*where $\sigma^2 = \prod_{i=1}^{l} \sigma_i^2$.*

Our proof is based on an inductive technique by conditioning on the pre-activations $\boldsymbol{f}^{(l)}$ of the previous layer while analyzing the pre-activations of the next layer. Due to space constraints, we defer the full proof of Thm 4.1 to the Appendix B.2. In the following, to give a flavor of our technique and highlight the great utility of the Meijer-G function, we present the base case as well as the key technical result (Lemma 4.2) to perform the inductive step to obtain the final statement.

**Base case ($L = 1$):** Here we restrict our attention to the first layer pre-activations which, given an input $\boldsymbol{x}$, are defined as:

$$f_k^{(1)}(\boldsymbol{x}) = \sum_{j=1}^{d_0} W_{jk}^{(1)} \cdot x_j. \tag{10}$$

Conditioned on the input, this is a sum of $d_0$ i.i.d. Gaussian random variables $W_{jk}^{(1)} \sim \mathcal{N}(0, \sigma_1^2)$, which is again Gaussian with mean zero and variance $\sigma_1^2 ||\boldsymbol{x}||^2$, i.e. $f_k^{(1)}(\boldsymbol{x}) \sim \mathcal{N}(0, \sigma_1^2 ||\boldsymbol{x}||^2) \stackrel{(d)}{=} \mathcal{N}(0, \sigma_1^2)$. where the last equality follows from $||\mathbf{x}|| = 1$. As we are conditioning on the input, the joint distribution of the first layer units is composed of $d_1$ independent Gaussians and hence

$$\boldsymbol{f}^{(1)}(\boldsymbol{x}) \sim \mathcal{N}(0, \sigma_1^2 \boldsymbol{I}). \tag{11}$$

Indeed, as anticipated from Thm. 4.1, the corresponding Meijer-G function encodes the Gaussian density:

$$G_{0,1}^{1,0}\left(\frac{||\boldsymbol{f}^{(1)}||^2}{2\sigma_1^2}\Big|0\right) = \exp\left(-\frac{||\boldsymbol{f}^{(1)}||^2}{2\sigma_1^2}\right). \tag{12}$$

**Induction Step:**   Conditioning on the previous pre-activations $\boldsymbol{f}^{(l-1)}$ brings us into a similar setting as in the base case because the resulting conditional distribution is again a Gaussian:

$$p(\boldsymbol{f}^{(l)}|\boldsymbol{f}^{(l-1)}) = \mathcal{N}(\mathbf{0}, \sigma_l^2||\boldsymbol{f}^{(l-1)}||_2^2). \tag{13}$$

In contrast to the fixed input $\boldsymbol{x}$, we now need to apply the law of total probability to integrate out the dependence on $\boldsymbol{f}^{(l-1)}$, leveraging the induction hypothesis for $p(\boldsymbol{f}^{(l-1)})$, to obtain the marginal distribution $p(\boldsymbol{f}^{(l)})$. This is where the Meijer-G function comes in handy as it can easily express such an integral. In combination with the closedness of the family under integration, this enables us to perform an inductive proof. We summarize this in the following Lemma:

**Lemma 4.2.** *Let $\boldsymbol{f}^l$ and $\boldsymbol{f}^{l-1}$ be $d_l$-dimensional and $d_{l-1}$-dimensional vectors, respectively, where $l > 1$, $l \in \mathbb{N}$. Let $\sigma_l^2 > 0$, $\tilde{\sigma}^2 > 0$ and $b_1, \ldots b_{l-1} \in \mathbb{R}$. Then the following integral:*

$$I := \int_{\mathbb{R}^{d_{l-1}}} p(\boldsymbol{f}^{(l)}|\boldsymbol{f}^{(l-1)})G_{0,l-1}^{l-1,0}\left(\frac{||\boldsymbol{f}^{(l-1)}||^2}{2^{l-1}\tilde{\sigma}^2}\Big|b_1,\ldots,b_{l-1}\right)d\boldsymbol{f}^{l-1}, \tag{14}$$

*can be expressed as:*

$$I = \frac{1}{C}G_{0,l}^{l,0}\left(\frac{||\boldsymbol{f}^{(l)}||^2}{2^l\sigma^2}\Big|0, \frac{1}{2}(d_{l-1}-d_l)+b_1, \ldots, \frac{1}{2}(d_{l-1}-d_l)+b_{l-1}\right), \tag{15}$$

*where $\sigma^2 = \sigma_l^2\tilde{\sigma}^2$, and $C \in \mathbb{R}$ is a constant available in closed-form.*

The proof largely relies on well-known integral identities involving the Meijer-G function. Again, due to its technical nature, we refer for the full proof and for the exact constant $C$ to the Appendix B.1 and B.6, respectively.

**ReLU Networks:**   In the previous paragraph we have computed the prior predictive distribution for a linear network with Gaussian weights. Now, we extend these results to ReLU networks. The proof technique used is very similar to its linear counterpart, the main difference stems from the need to decompose the distribution over active and inactive ReLU cells. As a consequence, the resulting density is a superposition of different Meijer-G functions, each associated with a different active (linear) subnetwork. This is presented in the following:

**Theorem 4.3.** *Suppose $l \geq 2$, and the input has dimension $d_0$. Define the multi-index set $\mathcal{R} = [d_1] \times \cdots \times [d_{l-1}]$ and introduce the vector $\boldsymbol{u^r} \in \mathbb{R}^{l-1}$ through its components $\boldsymbol{u}_i^r = \frac{1}{2}(r_i - d_l)$.*

$$p(\boldsymbol{f}_{ReLU}^{(l)}) = \sum_{\boldsymbol{r}\in\mathcal{R}} q_{\boldsymbol{r}} G_{0,l}^{l,0}\left(\frac{||\boldsymbol{f}_{ReLU}^{(l)}||^2}{2^l\sigma^2}\Big|0, \boldsymbol{u^r}\right) + q_0\delta(\boldsymbol{f}_{ReLU}^{(l)}), \tag{16}$$

*where $\sigma^2 = \prod_{i=1}^l \sigma_i^2$ and the individual weights are given by*

$$q_{\boldsymbol{r}} = \pi^{-\frac{d_l}{2}} 2^{-\frac{l}{2}d_l}(\sigma^2)^{-\frac{d_l}{2}} \prod_{i=1}^{l-1} \binom{d_i}{r_i} \frac{1}{2^{d_i}\Gamma\left(\frac{r_i}{2}\right)}, \tag{17}$$

*and*

$$q_0 = 1 - \prod_{i=1}^{l-1} \frac{2^{d_i}-1}{2^{d_i}} \tag{18}$$

We refer to Appendix C.2 for the detailed proof. Observe that sub-networks with the same number of active units per layer induce the same density. The theorem reflects this symmetry by not summing over all possible sub-networks but only over the representatives of each equivalence class, absorbing

the contribution of the other equivalent networks into $q_r$. Indeed, notice that $|\mathcal{R}| = \prod_{i=1}^{l-1} d_i$, which is exactly the number of equivalent sub-networks contained in the original network. Notice also that $q_r$ governs how much weight is assigned to the corresponding Meijer-G function. The normalization constant is incorporated into the weights $q_r$ since the Meijer-G function does not integrate to 1 (see Appendix B.4). As a result, the weights add up to the sum of the respective integration constants, and not to 1.

## 5 Analytic Insights into the Prior Predictive Distribution

Here we highlight how one can use the mathematical machinery of Meijer-G functions to derive interesting insights, relying on numerous mathematical results provided in the literature (Gradshteyn and Ryzik, 2013; Brychkov, 2008; Andrews, 2011). With this rich line of work at our disposal, we can easily move from the rather abstract but mathematically convenient world of Meijer-G functions to very concrete results. We demonstrate this by recovering and extending the NNGP results provided in Lee et al. (2018); Matthews et al. (2018). In particular, our analysis allows for simultaneous width and depth limits, showing how different limiting distributions emerge as a consequence of the growth with respect to $L$. Finally, we characterize the heavy-tailedness of the prior-predictive for any width, providing further evidence that deeper models induce distributions with heavier tails, as observed in Vladimirova et al. (2019). We hope that this work paves the way for further progress in understanding priors, leveraging this novel connection through the rich literature on Meijer-G functions.

### 5.1 Infinite Width: Recovering and Extending the NNGP for a Single Datapoint

We will start by giving an alternative proof in the linear case for the Gaussian behaviour emerging as the width of the network tends to infinity, recovering the results of Lee et al. (2018); Matthews et al. (2018) in the restricted setting of having just one fixed input $x$. We extend their results in the following ways:

1. We provide a convergence proof that is independent of the ordering of limits.
2. We characterize the distributions arising from a simultaneous infinite width and depth limit, considering different growth rates for depth $L$.

For ease of exposition, we focus on the equal width case, i.e. where $d_1 = \cdots = d_{l-1} = m$ with one output $d_L = 1$. To have well-defined limits, one has to resort to the so-called NTK parametrization (Jacot et al., 2018), which is achieved by setting the variances as $\sigma_1^2 = 1$ and $\sigma_i^2 = \frac{2}{m}$ for $i = 2, \ldots, l$. We summarize the result in the following:

**Theorem 5.1.** *Consider the distribution of the output* $p\left(f_{ReLU}^{(L)}\right)$, *as defined in Thm. 4.3. Denote* $X \sim \mathcal{N}(0,1)$, $Y \sim \mathcal{LN}(-\frac{5}{4}\gamma, \frac{5}{4}\gamma)$ *for* $X \perp Y$ *and the resulting normal log-normal mixture by* $Z = XY$, *for* $\gamma > 0$. *Let the depth grow as* $L = c + \gamma m^\beta$ *where* $\beta \geq 0$ *and* $c \in \mathbb{N}$ *fixed. Then it holds that*

$$\mathbb{E}\left[\left(f_{ReLU}^{(L)}\right)^{2k}\right] \xrightarrow{m\to\infty} \begin{cases} \mathbb{E}[X^{2k}] = (2k-1)!! & \text{if } \beta < 1 \\ \mathbb{E}[Z^{2k}] = e^{\frac{5}{2}\gamma k(k-1)}(2k-1)!! & \text{if } \beta = 1 \\ \infty & \text{if } \beta > 1 \end{cases}, \qquad (19)$$

*where* $(2k-1)!! = (2k-1)\ldots 3 \cdot 1$ *denotes the double factorial (by symmetry, odd moments are zero). Moreover, for* $\beta < 1$ *it holds that*

$$p(f_{ReLU}^{(L)}) \xrightarrow{d} X \quad \text{for } m \to \infty, \qquad (20)$$

*for any $k > 1$.*

The proof leverages the fact that we can easily calculate all the moments of $f_{ReLU}^{(L)}$ at finite width using well-known integral identities involving the Meijer-G function. For the limit, care has to be taken as the moments of $f_{ReLU}^{(L)}$ involve the moments of the Binomial distribution $\text{Bin}(m, \frac{1}{2})$ for which recursive but no analytic formulas are available. By carefully studying the leading coefficients of the

resulting polynomial, we can establish the result in Thm. 5.1.

This result is in stark contrast to Lee et al. (2018); Matthews et al. (2018) which cannot deal with a simultaneous depth limit due to the inductive nature of their proof, but can only describe the special case of fixed depth ($\gamma = 1, \beta = 0$). Notice that for depth growing asymptotically at a slower rate than width ($L = \gamma m^\beta$ for $\beta < 1$), we obtain the convergence in distribution since the Gaussian distribution is identified by its moments (See Theorem 30.2 in Billingsley (1986)). For $\beta = 1$, we can also prove convergence of the moments and identify them as arising from the normal log-normal mixture $Z$ (Yang, 2008). Unfortunately, this does not suffice to conclude convergence in distribution as there are no known results on the identifiability of $Z$. Empirical evidence however, presented in Figure 2 and Figure 3, suggests that the normal log-normal mixture captures the distribution to an excellent degree of fidelity. We demonstrate this through the following experiment. Note first that $Z \to \mathcal{N}(0, 1)$ in law as $\gamma \to 0$, which is expected, as for $\gamma \to 0$, we assume a fixed depth $L = c$. We can hence use this result in the non-asymptotic regime by setting $\gamma = \frac{L}{m}$, expecting to interpolate between the two distributions as we vary depth and width. We illustrate the empirical findings in Figure 2 and Figure 3. We find an astonishing match to the true distribution both in terms of CDF and PDF. Moreover, as expected, we recover the Gaussian distribution for small values of $\gamma$.

The decoupling of the moments into two separate factors gives insights into the role of width and depth on the shape of the distribution. The emergence of the log-normal factor in this infinite depth limit highlights how deeper networks encourage more heavy-tailed distribution, if not countered by a sufficient amount of width. This becomes even more drastic once depth outgrows width ($\beta > 1$), leading to divergence of all even moments. In the following section we will show that also in the finite width regime, heavy-tailedness becomes more pronounced as we increase the depth.

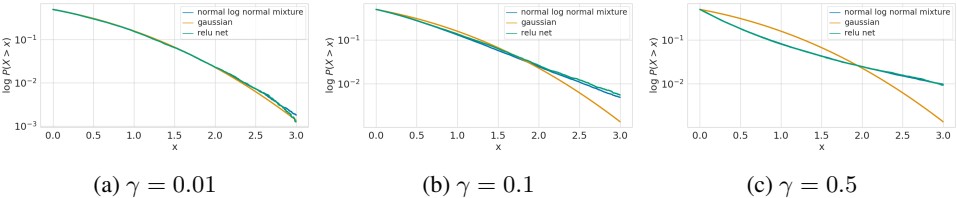

(a) $\gamma = 0.01$        (b) $\gamma = 0.1$        (c) $\gamma = 0.5$

Figure 2: **Convergence to normal log-normal mixture:** We display the CDFs of a ReLU network, standard Gaussian CDF and the normal log-normal mixture. The width is fixed to $100$ while depth is increased from $1$ to $50$ using the parameter $\gamma$. The neural network CDF is constructed empirically by drawing $10^4$ samples from the prior. Note that as depth increases, the CDF departs from the Gaussian but still follows the normal log-normal mixture.

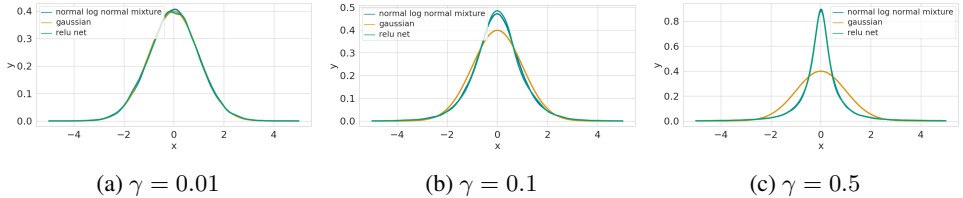

(a) $\gamma = 0.01$        (b) $\gamma = 0.1$        (c) $\gamma = 0.5$

Figure 3: **Convergence to normal log-normal mixture:** We show the density plots for the same setting as in Figure 2. Note that as depth increases, the density departs from the Gaussian but still follows the normal log-normal mixture.

## 5.2 Heavy-tailedness Increases with Depth

From the moments analysis, it is simple to recover a known fact about the prior distribution of neural networks, namely that deeper layers are increasingly heavy-tailed (Vladimirova et al., 2019). To see this, we can derive the kurtosis, a standard measure of tailedness of the distribution (Westfall, 2014), defined as:

$$\kappa := \frac{\mathbb{E}\left[ (X - \mathbb{E}[X])^4 \right]}{\mathbb{V}[X]^2}, \tag{21}$$

where $X$ is a univariate random variable with finite fourth moment. We can calculate the kurtosis of a ReLU network analytically at any width, relying on closed-form results for the lower order moments of the Binomial distribution. We outline the exact calculation in the Appendix C and state the resulting expression here:

$$\kappa_{\text{ReLU}}(m, L) = 3 \left( \frac{m+5}{m} \right)^{L-1}. \tag{22}$$

Note how the kurtosis increases with depth $L$ and decreases with the width $m$, highlighting once again the opposite roles those two parameters take regarding the shape of the prior. As expected, for fixed depth $L$, the kurtosis converges to 3, i.e. $\lim_{m \to \infty} \kappa_{\text{ReLU}}(m, L) = 3$, which is the kurtosis of a standard Gaussian variable $\mathcal{N}(0, 1)$. For the simultaneous limit $L = \gamma m$, we find a value of $3e^{5\gamma}$, which exactly matches with the expression derived in Thm. 5.1:

$$\lim_{m \to \infty} \kappa_{\text{ReLU}}(m, \gamma m) = \frac{\mathbb{E}[Z^4]}{\mathbb{E}[Z^2]^2} = \frac{e^{\frac{5}{2}\gamma^2} 3!!}{e^0 1!!} = 3e^{5\gamma}. \tag{23}$$

## 6 Prior Design

In Section 5.2, we have already outlined how the choice of architecture influences the heavy-tailedness of the distribution. If a Gaussian-like output is desired, the architecture should be designed in such a way that the width $m$ significantly exceeds the depth $L$ (small $\gamma$), while heavy-tailed predictive priors can be achieved by considering regimes where $L$ exceeds $m$ (big $\gamma$). As we will see shortly after in this section, another consequence of our analysis is that we can now directly work with the variance in function space, instead of implicitly tuning it by changing the variances at each layer. In this way, the variance over the weights has a clear interpretation in terms of predictive variance, making it easier for the deep learning practitioner to take an informed decision when designing a prior for BNNs. This type of variance analysis has previously been used to devise better initialization schemes for neural networks coined "He-initialization" (He et al., 2015) in the Gaussian case and "Xavier initialization" (Glorot and Bengio, 2010) for uniform weight initializations. Therefore, we coin the resulting prior *Generalized He-prior*.

**Predictive variance:** We consider the parametrization $\sigma_1^2 = 1$ and $\sigma_i^2 = \frac{2t_i^2}{m}$, which, as we show in Appendix C, leads to the following predictive variance:

$$\mathbb{V}[Z] = \sigma_1^2 \prod_{i=2}^{l} t_i^2. \tag{24}$$

Suppose we want a desired output variance $\sigma^2$. This can be achieved as follows: let $a_i$, with $i = 2, \ldots, l$, be $l-1$ coefficients such that $\sum_{i=1}^{l-1} a_i = l-1$. Then choose $t_i^2 = \left( \sigma^2 \right)^{\frac{a_i}{l-1}}$, implying that the layer variances $\sigma_i^2$ are given as

$$\sigma_i^2 = \frac{2}{m} \left( \sigma^2 \right)^{\frac{a_i}{l-1}}, \tag{25}$$

He-priors correspond to the special case where $\sigma^2 = 1$ and $a_i = 1$, $i = 1, \ldots, l-1$, while a standard Gaussian $N(0, 1)$ prior results in $\sigma^2 = \left( \frac{m}{2} \right)^{l-1}$, $\sigma_1^2 = 1$, and $a_i = 1$, $i = 1, \ldots, l-1$. Note how "far" He-priors can be from standard Gaussian for relatively deep and wide neural networks. While rather innocent-looking, using a standard Gaussian $\mathcal{N}(0, 1)$ prior can lead to an extremely high output variance.

Combining with the previous insights, practitioners can now choose a desired output variance along with a desired level of heavy-tailedness in a controlled manner by specifying the architecture, i.e. the width $m$ and the depth $L$.

## 7 Discussion

Our work sheds light on the shape of the prior predictive distribution arising from imposing a Gaussian distribution on the weights. Leveraging the machinery of Meijer-G functions, we characterized the

density of the output in the finite width regime and derived analytic insights into its properties such as moments and heavy-tailedness. An extension to the stochastic process setting in the spirit of Lee et al. (2018); Matthews et al. (2018) as well as to convolutional architectures (Novak et al., 2019) could bring theory even closer to practice and we expect similar results to also hold in those cases. This is however beyond the scope of our work and we leave it as future work.

Our technique enabled us to extend the NNGP framework to infinite depth, discovering how in the more general case, the resulting distribution shares the same moments as a normal log-normal distribution. This allowed us to disentangle the roles of width and depth, where the former induces a Gaussian-like distribution while the latter encourages heavier tails. Empirically, we found that the normal log-normal mixture provides an excellent fit to the true distribution even in the non-asymptotic setting, capturing both the cumulative and probability density function to a very high degree of accuracy. This surprising observation begs further theoretical and empirical investigations. In particular, discovering a suitable stochastic process incorporating the normal log-normal mixture, could lend further insights into the inner workings of neural networks. Moreover, the role of heavy-tailedness regarding generalization is very intriguing, potentially being an important reason underlying the gap between infinite width networks and their finite counterparts. This is also in-line with recent empirical works on priors, suggesting that heavy-tailed distributions can increase the performance significantly (Fortuin et al., 2021).

Using these insights, we described how one can choose a fixed prior variance directly in the output space along with a desired level of heavy-tailedness resulting from the choice of the architecture. We leave it as future work to also consider higher moments to give more nuanced control over the resulting prior in the function space. Finally, we hope that the introduction of the Meijer-G function sparks more theoretical research on BNN priors and their implied inductive biases in function space.

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
