# Appendix

## Table of Contents

## A  Properties of Meijer-G function

Here we describe the properties of Meijer-G functions which we will use extensively in the following.

The first result concerns the Mellin transform of the Meijer-G function, which will be the key to solve the integrals that we will face later.

**Proposition A.1** ( Mellin transform of the Meijer G function)**.**

$$\int_0^\infty x^{s-1} G_{p,q}^{m,n}(wx) = w^{-s} \frac{\prod_{j=1}^m \Gamma(b_j + s) \prod_{j=1}^n \Gamma(1 - a_j - s)}{\prod_{j=m+1}^q \Gamma(1 - b_j - s) \prod_{j=n+1}^p \Gamma(a_j + s)}. \tag{26}$$

*Proof.* See Chapter 3.2 and 2.3 of Mathai and Saxena (2006). Conditions of validity: for the class of Meijer-G functions that we consider here ($p$, $n = 0$, $m = q = l$ and the coefficients are all real) the Mellin transform exists (see Mathai and Saxena (2006), Section 2.3.1). $\qquad\square$

To establish the base case $m = 1$, we need the following results.

**Proposition A.2.** *The following identities hold:*

- $\exp(z) = G_{0,1}^{1,0} \left( -z \Big| 0 \right).$

- *multiplication by power property:* $z^d G_{p,q}^{m,n} \left( z \Big|_{b_1\ \dots\ b_q}^{a_1\ \dots\ a_p} \right) = G_{p,q}^{m,n} \left( z \Big|_{b_1+d\ \dots\ b_q+d}^{a_1+d\ \dots\ a_p+d} \right).$

*Proof.* See Chapter 2.6 of Mathai and Saxena (2006) for the first identity. The last property follows directly from the definition of Meijer-G function. $\qquad\square$

To perform the inductive step, we will encounter the following integral, that can be expressed in terms of the Meijer-G function.

**Proposition A.3.**

$$\int_1^\infty x^{-\rho}(x-1)^{\sigma-1}G_{p,q}^{m,n}\left(\alpha x\Big|\begin{smallmatrix}a_1 & \cdots & a_p\\ b_1 & \cdots & b_q\end{smallmatrix}\right)dx = \Gamma(\sigma)G_{p+1,q+1}^{m+1,n}\left(\alpha\Big|\begin{smallmatrix}a_1 & \cdots & a_p & \rho\\ \rho-\sigma & b_1 & \cdots & b_q\end{smallmatrix}\right). \quad (27)$$

The conditions of validity for the class of Meijer-G that we consider here are again satisfied (see Appendix B of Stojanac et al. (2017)).

# B  Proof for Linear Networks and Derivation of their Moments

Here we collect all the results regarding linear networks, establishing the relevant technical Lemmas to derive the density and calculate the moments of the resulting distribution.

**Lemma B.1.** *The units of any layer are uncorrelated, i.e.*

$$Cov\left(f_k^{(l)}, f_{k'}^{(l)}\right) = 0, \quad (28)$$

*for all layers $l$, and for all $k, k' \in [d_l]$*

*Proof.*

$$Cov\left(f_k^{(l)}, f_{k'}^{(l)}\right) = Cov\left(\sum_{j=1}^{d_{l-1}} f_j^{(l-1)}W_{jk}^{(l)}, \sum_{j'=1}^{d_{l-1}} f_{j'}^{(l-1)}W_{j'k'}^{(l)}\right) \quad (29)$$

$$= \mathbb{E}\left[\sum_{j=1}^{d_{l-1}} f_j^{(l-1)}W_{jk}^{(l)} \sum_{j'=1}^{d_{l-1}} f_{j'}^{(l-1)}W_{j'k'}^{(l)}\right] \quad (30)$$

$$= \sum_{j=1}^{d_{l-1}}\sum_{j'=1}^{d_{l-1}} \mathbb{E}\left[f_j^{(l-1)}W_{jk}^{(l)}f_{j'}^{(l-1)}W_{j'k'}^{(l)}\right] \quad (31)$$

$$= \sum_{j=1}^{d_{l-1}}\sum_{j'=1}^{d_{l-1}} \mathbb{E}\left[f_j^{(l-1)}f_{j'}^{(l-1)}\right]\mathbb{E}\left[W_{jk}^{(l)}\right]\mathbb{E}\left[W_{j'k'}^{(l)}\right] \quad (32)$$

$$= 0. \quad (33)$$

$\square$

However, they are **not** independent, but only conditionally independent given the previous later's units. As a remark, note that as $d_1 \to \infty$, the units $f_k^{(2)}$ approach a Gaussian distribution, for which uncorrelation implies independence.

## B.1  Main technical Lemma for induction

Here we prove the main technical Lemma (Lemma 4.2) that allows us to perform the inductive step.

**Lemma.** *Let $\boldsymbol{f}^l$ and $\boldsymbol{f}^{l-1}$ be a $d_l$-dimensional and a $d_{l-1}$-dimensional vectors, respectively. Let $\sigma_w^2 > 0$, $\tilde{\sigma}^2 > 0$ be two variance parameters, and $b_1, \ldots b_{l-1} \in \mathbb{R}$. Then the following integral:*

$$I := \int_{\mathbb{R}^{d_{l-1}}} \frac{1}{(||\boldsymbol{f}^{(l-1)}||^2)^{\frac{d_l}{2}}} e^{-\frac{||\boldsymbol{f}^{(l)}||^2}{2\sigma_w^2 ||\boldsymbol{f}^{(l-1)}||^2}} G_{0,l-1}^{l-1,0} \left( \frac{||\boldsymbol{f}^{(l-1)}||^2}{2^{l-1}\tilde{\sigma}^2} \middle| b_1, \ldots, b_{l-1} \right) d\boldsymbol{f}^{l-1}, \quad (34)$$

*has solution:*

$$I = C G_{0,l}^{l,0} \left( \frac{||\boldsymbol{f}^{(l)}||^2}{2^l\sigma^2} \middle| 0, \frac{1}{2}(d_{l-1} - d_l) + b_1, \ldots, \frac{1}{2}(d_{l-1} - d_l) + b_{l-1} \right), \quad (35)$$

*where $\sigma^2 := \sigma_w^2\tilde{\sigma}^2$, and $C := \frac{1}{2}\tilde{C}2^{\frac{1}{2}(d_{l-1}-d_l)(l-1)}\tilde{\sigma}^{(d_{l-1}-d_l)}$, where $\tilde{C}$ is a constant that depends only on $d_{l-1}$.*

*Proof.* The proof is based in two steps: in the first steps, we will write the integral in hyper-spherical coordinates. In the second step, we will apply a useful substitution and the properties of the Meijer-G function to solve the integral.

**1. Hyper-spherical coordinates** Apply the following substitution:

$$f_1^{(l-1)} = r \cos(\gamma_1) \quad (36)$$
$$f_2^{(l-1)} = r \sin(\gamma_1) \cos(\gamma_2) \quad (37)$$
$$\cdots \quad (38)$$
$$f_{d_{l-1}}^{(l-1)} = r \sin(\gamma_1) \cdots \sin(\gamma_{d_{l-1}-1}), \quad (39)$$

where $r \in \mathbb{R}_{\geq 0}$ is the radius and $\gamma_1, \ldots \gamma_{d_{l-1}-2} \in [0, \pi]$ and $\gamma_{d_{l-1}-1} \in [0, 2\pi]$. The Jacobian is:

$$J_n = \begin{pmatrix} \cos(\gamma_1) & -r\sin(\gamma_1) & 0 & 0 & \cdots & 0 \\ \sin(\gamma_1)\cos(\gamma_2) & r\cos(\gamma_1)\cos(\gamma_2) & -r\sin(\gamma_1)\sin(\gamma_2) & 0 & \cdots & 0 \\ \cdots & & & & \cdots & \\ r\sin(\gamma_1)\cdots\sin(\gamma_{d_{l-1}-1}) & & & & \cdots & r\sin(\gamma_1)\cdots\cos(\gamma_{d_{l-1}-1}) \end{pmatrix}$$
$$(40)$$

, where it can be shown that its determinant is:

$$|J_n| = r^{d_{l-1}-1} \sin^{d_{l-1}-2}(\gamma_1) \sin^{d_{l-1}-3}(\gamma_2) \cdots \sin(\gamma_{d_{l-1}-2}). \quad (41)$$

Therefore:

$$\prod_{i=1}^{d_{l-1}} df_i^{(l-1)} = r^{d_{l-1}-1} \sin^{d_{l-1}-2}(\gamma_1) \sin^{d_{l-1}-3}(\gamma_2) \cdots \sin(\gamma_{d_{l-1}-2}) dr d\gamma_1 \cdots d\gamma_{d_{l-1}-1}. \quad (42)$$

By noting that the integral we are trying to solve depends only on $||\boldsymbol{f}^{(l-1)}||^2 = r^2$, we have that the density is, up to a normalization constant independent of $\boldsymbol{f}^{(l)}$:

$$I = \tilde{C} \int r^{d_{l-1}-d_l-1} e^{-\frac{||\boldsymbol{f}^{(l)}||^2}{2\sigma_w^2 r^2}} G_{0,l-1}^{l-1,0} \left( \frac{r^2}{2^{l-1}\tilde{\sigma}^2} \middle| b_1, \ldots, b_{l-1} \right) dr \quad (43)$$

$$= \tilde{C} \int r^{d_{l-1}-d_l-1} G_{0,1}^{1,0} \left( \frac{||\boldsymbol{f}^{(l)}||^2}{2\sigma_w^2 r^2} \middle| 0 \right) G_{0,l-1}^{l-1,0} \left( \frac{r^2}{2^{l-1}\tilde{\sigma}^2} \middle| b_1, \ldots, b_{l-1} \right) dr, \quad (44)$$

where we call $\tilde{C}$ the angular constant due to the integration of the angle-related terms (that do not depend on $r$, but only on $d_{l-1}$). We compute the angular constant in Lemma B.4. In the last step we have applied the identity between the exponential function and the Meijer-G function as in Proposition A.2.

**2: Substitution and Meijer-G properties** Defining $d = \frac{1}{2}(d_{l-1} - d_l)$, and applying the substitution $x = \frac{r^2}{2^{l-1}\tilde{\sigma}^2}$:

$$I = \tilde{C} \int (2^{\frac{l-1}{2}}\tilde{\sigma}x^{\frac{1}{2}})^{2d-1} G_{0,1}^{1,0}\left(\frac{||\boldsymbol{f}^{(l)}||^2}{2^l\sigma_w^2\tilde{\sigma}^2 x}\bigg|0\right) G_{0,l-1}^{l-1,0}\left(x\bigg|b_1,\ldots,b_{l-1}\right) 2^{\frac{l-3}{2}}\tilde{\sigma}x^{-\frac{1}{2}}dx \quad (45)$$

$$= \frac{1}{2}\tilde{C}2^{d(l-1)}\tilde{\sigma}^{2d} \int x^{d-1} G_{0,1}^{1,0}\left(\frac{||\boldsymbol{f}^{(l)}||^2}{2^l\sigma_w^2\tilde{\sigma}^2 x}\bigg|0\right) G_{0,l-1}^{l-1,0}\left(x\bigg|b_1,\ldots,b_{l-1}\right) dx. \quad (46)$$

Defining $\sigma^2 = \sigma_w^2\tilde{\sigma}^2$, $a^2 := \frac{||\boldsymbol{f}^{(l)}||^2}{2^l\sigma^2}$ and $C := \frac{1}{2}\tilde{C}2^{d(l-1)}\tilde{\sigma}^{2d}$, and expanding the $G_{0,1}^{1,0}$ term according to the definition, we get:

$$I = C \int x^{d-1}\frac{1}{2\pi i}\int \Gamma(s)\left(\frac{a^2}{x}\right)^{-s} G_{0,l-1}^{l-1,0}\left(x\bigg|b_1,\ldots,b_{l-1}\right) ds dx \quad (47)$$

$$= C\frac{1}{2\pi i}\int \Gamma(s)a^{-2s}\int x^{s+d-1}G_{0,l-1}^{l-1,0}\left(x\bigg|b_1,\ldots,b_{l-1}\right) dx ds. \quad (48)$$

where we can change the order of integration due to the fact that the integrand is positive in the integration region (Tonelli's theorem). Now by using Proposition A.1, the inner integral has the following solution:

$$\int x^{s+d-1}G_{0,l-1}^{l-1,0}\left(x\bigg|b_1,\ldots b_{l-1}\right) dx = \prod_{i=1}^{l-1}\Gamma(d+b_i+s) \quad (49)$$

$$= \prod_{i=1}^{l-1}\Gamma\left(\frac{1}{2}(d_{l-1}-d_l)+b_i+s\right). \quad (50)$$

Therefore we can conclude that:

$$I = C\frac{1}{2\pi i}\int \Gamma(s)\prod_{i=1}^{l-1}\Gamma(\frac{1}{2}(d_{l-1}-d_l)+b_i+s)a^{-2s}ds \quad (51)$$

$$= CG_{0,l}^{l,0}\left(a^2\bigg|0,\frac{1}{2}(d_{l-1}-d_l)+b_1,\ldots,\frac{1}{2}(d_{l-1}-d_l)+b_{l-1}\right) \quad (52)$$

$$= CG_{0,l}^{l,0}\left(\frac{||\boldsymbol{f}^{(l)}||^2}{2^l\sigma^2}\bigg|0,\frac{1}{2}(d_{l-1}-d_l)+b_1,\ldots,\frac{1}{2}(d_{l-1}-d_l)+b_{l-1}\right), \quad (53)$$

where we have simply applied the definition of the Meijer-G function.

$\square$

## B.2 Probability Density Function for linear networks

We proof the result on the probability density function for a linear network in the following.

**Theorem.** *Suppose $l \geq 1$, and the input has dimension $d_0$. Then, the joint marginal density of the random vector $\boldsymbol{f}^{(l)}$ (i.e. the density of the $l$-th layer pre-activations) is proportional to:*

$$p(\boldsymbol{f}^{(l)}) \propto G_{0,l}^{l,0}\left(\frac{||\boldsymbol{f}^{(l)}||^2}{2^l\sigma^2}\bigg|0,\frac{1}{2}(d_1-d_l),\ldots,\frac{1}{2}(d_{l-1}-d_l)\right), \quad (54)$$

*where $\sigma^2 = \prod_{i=1}^{l}\sigma_i^2$.*

*Proof.* We proof by induction. For the base case, consider $l = 1$. We have shown that

$$\boldsymbol{f}^{(1)} \sim \mathcal{N}(0,\sigma_1^2 I). \quad (55)$$

Therefore we can re-write its density as:

$$p(f^{(1)}, \ldots f^{(d)}) = \frac{1}{(2\pi\sigma_1^2)^{\frac{d_1}{2}}} \exp\left(-\frac{||\boldsymbol{f}^{(1)}||^2}{2\sigma_1^2}\right) \tag{56}$$

$$= \frac{1}{(2\pi\sigma_1^2)^{\frac{d_1}{2}}} G_{0,1}^{1,0}\left(\frac{||\boldsymbol{f}^{(1)}||^2}{2\sigma_1^2}\bigg| 0\right), \tag{57}$$

where we have used the identity between the exponential function and the Meijer-G function (Proposition A.2).

Now let $\tilde{\sigma}^2 = \prod_{i=1}^{l-1} \sigma_i^2$. Assume that

$$p(f_1^{(l)}, \ldots, f_{d_{l-1}}^{(l-1)}) \propto G_{0,l-1}^{l-1,0}\left(\frac{||\boldsymbol{f}^{(l-1)}||^2}{2^{l-1}\tilde{\sigma}^2}\bigg| 0, \frac{1}{2}(d_1 - d_{l-1}), \ldots, \frac{1}{2}(d_{l-2} - d_{l-1})\right). \tag{58}$$

Now we can use the fact that the the units of the $l$-th layer are conditionally independent given the previous' layer units. Furthermore the conditional distribution is Gaussian due to the fact that the weights are i.i.d Gaussian. Therefore we can write:

$$p(f_1^{(l)}, \ldots f_{d_l}^{(l)}) = \int_{\mathbb{R}^{d_{l-1}}} p(f_1^{(l)}, \ldots f_{d_l}^{(l)} | f_1^{(l-1)}, \ldots f_{d_1}^{(l-1)}) p(f_1^{(l-1)}, \ldots f_{d_1}^{(l-1)}) d\boldsymbol{f}^{(l-1)} \tag{59}$$

$$\propto \int_{\mathbb{R}^{d_{l-1}}} \frac{1}{(2\pi||\boldsymbol{f}^{(l-1)}||^2)^{\frac{d_l}{2}}} e^{-\frac{||\boldsymbol{f}^{(l)}||^2}{2\sigma_l^2||\boldsymbol{f}^{(l-1)}||^2}} \tag{60}$$

$$G_{0,l-1}^{l-1,0}\left(\frac{||\boldsymbol{f}^{(l-1)}||^2}{2^{l-1}\tilde{\sigma}^2}\bigg| 0, \frac{1}{2}(d_1 - d_{l-1}), \ldots, \frac{1}{2}(d_{l-2} - d_{l-1})\right) d\boldsymbol{f}^{(l-1)}. \tag{61}$$

In the first step we have marginalized out the units of the $l-1$ layer, and applied the product rule of probabilities. In the second step we have applied the induction hypothesis.

The integral is in the form of Lemma 4.2. For the coefficients of the Meijer-G function $b_2 = \frac{1}{2}(d_1 - d_{l-1}), \ldots, b_{l-1} = \frac{1}{2}(d_{l-2} - d_{l-1})$, note that:

$$\frac{1}{2}(d_i - d_{l-1}) + \frac{1}{2}(d_{l-1} - d_l) = \frac{1}{2}(d_i - d_l) \tag{62}$$

holds for all $i \in [d_{l-2}]$ and clearly $b_1 + \frac{1}{2}(d_{l-1} - d_l) = \frac{1}{2}(d_{l-1} - d_l)$ as $b_1 = 0$ in our case. Therefore by Lemma 4.2 we can conclude that:

$$p(f_1^{(l)}, \ldots f_{d_l}^{(l)}) \propto G_{0,l}^{l,0}\left(\frac{||\boldsymbol{f}^{(l)}||^2}{2^l\sigma^2}\bigg| 0, \frac{1}{2}(d_1 - d_l) \ldots, \frac{1}{2}(d_{l-1} - d_l)\right). \tag{63}$$

$\square$

## B.3 CDF of prior predictive

We also derive the CDF of the linear network in the following theorem and proceed to prove it.

**Theorem B.2** (CDF of prior predictive). *Let $f^l$ be the output of a of a linear network of $l$ layers. We assume the final layer is one dimensional. Then the the cdf is $F_l(t) := 1 - P(f^l > t)$, $t > 0$. We have that*

$$P(f^l > t) = \frac{t}{2C} G_{1,l+1}^{l+1,0}\left(\omega t^2 \bigg| \begin{matrix} \frac{1}{2} \\ -\frac{1}{2} \ 0 \ b_1 \ \ldots \ b_{l-1} \end{matrix}\right), \tag{64}$$

*where $b_i = \frac{1}{2}(d_i - 1)$, $i \in [l-1]$, $C$ is the normalization constant and $\omega = \frac{1}{2^l\sigma^2}$.*

*Proof.* Let $X = f^l$.

$$P(X > t) = \frac{1}{C} \int_t^\infty G_{0,l}^{l,0} \left( \omega x^2 \Big| 0, b_1, \ldots, b_{l-1} \right) dx \tag{65}$$

$$= \frac{t}{2C} \int_1^\infty y^{-\frac{1}{2}} G_{0,l}^{l,0} \left( \omega t^2 y \Big| 0, b_1, \ldots, b_{l-1} \right) dx \tag{66}$$

$$= \frac{t}{2C} G_{1,l+1}^{l+1,0} \left( \omega t^2 \Big| \begin{smallmatrix} \frac{1}{2} \\ -\frac{1}{2} \ 0 \ b_1 \ \ldots \ b_{l-1} \end{smallmatrix} \right), \tag{67}$$

where in the first step we have used the result of 4.1, in the second step we have applied the substitution $y = \frac{x^2}{t^2}$, and in the last step we have used Equation 27 with $\rho = \frac{1}{2}$, $\sigma = 1$ and $\alpha = \omega t^2$ $\qquad\square$

## B.4 Resulting Moments for Linear Networks

Define $\omega = \frac{1}{2^l \sigma^2}$. Denote by $\tilde{p}$ the unnormalized measure and define the random variable

$$Z = ||\boldsymbol{f}^{(l)}||_2^2. \tag{68}$$

We are interested in the k-th moment of $Z$. Using spherical coordinates and the properties of the Meijer-G function in a similar way as the proofs above, we get:

$$\mathbb{E}\left[Z^k\right] = \frac{1}{C} \int_{\mathbb{R}^m} ||\boldsymbol{z}||_2^{2k} G_{0,l}^{l,0} \left( \omega ||\boldsymbol{z}||_2^2 \Big| 0, \frac{1}{2}(d_1 - d_l), \ldots, \frac{1}{2}(d_{l-1} - d_l) \right) d\boldsymbol{z} \tag{69}$$

$$= \frac{\tilde{C}_l}{C} \int_0^\infty r^{2k+d_l-1} G_{0,l}^{l,0} \left( \omega r^2 \Big| 0, \frac{1}{2}(d_1 - d_l), \ldots, \frac{1}{2}(d_{l-1} - d_l) \right) dr \tag{70}$$

$$= \frac{\tilde{C}_l}{2C} \int_0^\infty x^{k+\frac{d_l}{2}-1} G_{0,l}^{l,0} \left( \omega x \Big| 0, \frac{1}{2}(d_1 - d_l), \ldots, \frac{1}{2}(d_{l-1} - d_l) \right) dx \tag{71}$$

$$= \frac{\tilde{C}_l}{2C} \omega^{-k-\frac{d_l}{2}} \prod_{i=1}^l \Gamma\left( \frac{d_i}{2} + k \right) \tag{72}$$

$$= \omega^{-k} \frac{\prod_{i=1}^l \Gamma\left( \frac{d_i}{2} + k \right)}{\prod_{i=1}^l \Gamma\left( \frac{d_i}{2} \right)} \tag{73}$$

$$= (2^l \sigma^2)^k \prod_{i=1}^l \frac{\Gamma\left( \frac{d_i}{2} + k \right)}{\Gamma\left( \frac{d_i}{2} \right)}. \tag{74}$$

Note that it can be equivalently written as:

$$\mathbb{E}\left[Z^k\right] = (2^l \sigma^2)^{k-1} \prod_{i=1}^l \left( \frac{\Gamma\left( \frac{d_i}{2} + k - 1 \right)}{\Gamma\left( \frac{d_i}{2} \right)} \right) (2^l \sigma^2) \prod_{i=1}^l \left( \frac{d_i}{2} + k - 1 \right) \tag{75}$$

$$= \mathbb{E}\left[Z^{k-1}\right] \sigma^2 \prod_{i=1}^l (d_i + 2(k-1)), \tag{76}$$

so the kurtosis is:

$$\kappa = \frac{\prod_{i=1}^l (d_i + 2(2-1))}{\prod_{i=1}^l d_i} = \prod_{i=1}^l \frac{d_i + 2}{d_i} \tag{77}$$

If $d_1 = \cdots = d_{l-1} = m$, and $d_l = 1$. For instance the variance ($k = 1$) is[1]:

$$(2^l \sigma^2) \left( \frac{\Gamma(\frac{m}{2} + 1)}{\Gamma(\frac{m}{2})} \right)^{l-1} \frac{\Gamma(1 + \frac{1}{2})}{\Gamma(\frac{1}{2})} = (2^l \sigma^2) \frac{m^{l-1}}{2^{l-1}} \frac{1}{2} = \sigma^2 m^{l-1}. \tag{78}$$

---

[1]By symmetry, all the odd moments are zero

## B.5 Infinite width and depth limit

We also present the infinite-width and infinite-depth result for the linear case. Due to the linear nature, the proof simplifies significantly compared to the ReLU case.

---

**Lemma B.3.** *Consider the distribution of the output $p(f^{(L)})$, as defined in Thm. 4.1. Denote $X \sim \mathcal{N}(0,1)$, $Y \sim \mathcal{LN}(-\frac{\gamma}{2}, \frac{\gamma}{2})$ and the normal log-normal mixture $Z = XY$.*

*For fixed depth $L \in \mathbb{N}$, under NTK parametrization , it holds that*

$$p^m(f^{(L)}) \xrightarrow{d} X \quad for\ m \to \infty \tag{79}$$

*In contrast, for growing depth $L = \gamma m$, we have the following convergence of the moments*

$$\mathbb{E}\left[\left(f^{(L)}\right)^{2k}\right] \xrightarrow{m \to \infty} \mathbb{E}[Z^{2k}] = e^{\gamma k(k-1)}(2k-1)!! \tag{80}$$

*where $(2k-1)!! = (2k-1)\ldots 3 \cdot 1$ denotes the double factorial.*

---

*Proof.* Recall that the moments of $||\boldsymbol{f}^{(l)}||_2$ are given by

$$\mathbb{E}\left[||\boldsymbol{f}^{(l)}||_2^{2k}\right] = (2^l \sigma^2)^k \prod_{i=1}^{l} \frac{\Gamma\left(\frac{d_i}{2} + k\right)}{\Gamma\left(\frac{d_i}{2}\right)}, \tag{81}$$

where $\sigma^2 = \prod_{i=1}^{l} \sigma_i^2$. Assuming $d_1 = \ldots d_{l-1} = m$ and $d_l = 1$ and the NTK parametrization $\sigma_1^2 = 1$ and $\sigma_2^2 = \cdots = \sigma_l^2 = \frac{1}{m}$ simplifies this to

$$\mathbb{E}\left[\left(f^{(l)}\right)^{2k}\right] = \left(\frac{2^l}{m^{l-1}}\right)^k \frac{\Gamma(\frac{m}{2} + k)^{l-1}}{\Gamma(\frac{m}{2})^{l-1}} \frac{\Gamma(\frac{1}{2} + k)}{\Gamma(\frac{1}{2})} \tag{82}$$

$$= \left(\frac{2^l}{m^{l-1}}\right)^k \frac{\Gamma(\frac{m}{2} + k)^{l-1}}{\Gamma(\frac{m}{2})^{l-1}} 2^{-k}(2k-1)!! \tag{83}$$

$$= \left(\frac{2^k}{m^k} \frac{\Gamma(\frac{m}{2} + k)}{\Gamma(\frac{m}{2})}\right)^{l-1} (2k-1)!! \tag{84}$$

$$= \left(\frac{2^k}{m^k} \left(\frac{m}{2} + k - 1\right) \ldots \left(\frac{m}{2} + 1\right) \frac{m}{2}\right)^{l-1} (2k-1)!!. \tag{85}$$

Define the $k$-th order polynomial $p(m) = \left(\frac{m}{2} + k - 1\right) \ldots \left(\frac{m}{2} + 1\right) \frac{m}{2}$. Denote its coefficients by $\alpha_i$ for $i = 1, \ldots, k$. We know that $\alpha_k = \frac{1}{2^k}$ and from Lemma C.3 that

$$\alpha_{k-1} = \frac{1}{2^{k-1}} \sum_{i=1}^{k} (k-i) = \frac{1}{2^{k-1}} \left(k^2 - \frac{k(k+1)}{2}\right) = \frac{k^2 - k}{2^k}. \tag{86}$$

Assuming constant depth, performing the division by $m^k$ thus leads to

$$\left(2^k \left(\alpha_k + \alpha_{k-1} \frac{1}{m} + \cdots + \mathcal{O}\left(\frac{1}{m^2}\right)\right)\right)^{l-1} (2k-1)!!$$

$$= \left(1 + \frac{k(k-1)}{m} + \mathcal{O}\left(\frac{1}{m^2}\right)\right)^{l-1} (2k-1)!! \tag{87}$$

$$= \left(1 + \frac{(l-2)\left((k-1)k\right)}{m} + \mathcal{O}\left(\frac{1}{m^2}\right)\right)(2k-1)!!.$$

Now we can easily see that

$$\mathbb{E}\left[\left(f^{(l)}\right)^{2k}\right] \xrightarrow{m \to \infty} (2k-1)!!. \tag{88}$$

Recall that for $X \sim \mathcal{N}(0,1)$ we have the same moments $\forall k \in \mathbb{N}$: $\mathbb{E}\left[X^{2k}\right] = (2k-1)!!$, whereas the odd moments vanish for both distributions due to symmetry. The convergence of the moments, due to Billingsley (1986) and the identifiability of the Gaussian distribution implies convergence in distribution.

On the other hand, if we assume that depth grows proportional to width, i.e. $l - 1 = \gamma m$ for $\gamma > 0$, we arrive at a different limit given by

$$\left(1 + \frac{k(k-1)}{m} + \mathcal{O}\left(\frac{1}{m^2}\right)\right)^{\gamma m}(2k-1)!! \xrightarrow{m \to \infty} e^{\gamma k(k-1)}(2k-1)!!. \tag{89}$$

Consider the random variable $Z = XY$ where $X \sim \mathcal{N}(0,1)$ and $Y \sim \mathcal{LN}(s, t^2)$ are two independent variables. For $k \in \mathbb{N}$, we can compute the moments as

$$\mathbb{E}\left[Z^n\right] = \mathbb{E}\left[X^n Y^n\right] = \mathbb{E}\left[X^n\right]\mathbb{E}\left[Y^n\right] = \begin{cases} 0 & \text{n odd} \\ (2k-1)!! e^{2ks + 2k^2 t^2} & \text{n = 2k} \end{cases} \tag{90}$$

Choosing $s = -\frac{\gamma}{2}$ and $t^2 = \frac{\gamma}{2}$ hence recovers the moments exactly. $\qquad\square$

## B.6 Normalization Constant and Angular Constant

We complete the picture by calculating the normalization constant of the resulting distribution.

**Lemma B.4** (normalization constant). *Under the conditions of Theorem 4.1, the normalization constant $C$ for the density of the l-th layer can be computed as:*

$$C = \frac{1}{2}\tilde{C}_l\left(\frac{1}{2^l\sigma^2}\right)^{-\frac{d_l}{2}}\prod_{i=1}^{l}\Gamma\left(\frac{d_i}{2}\right), \tag{91}$$

*or, expanding $\tilde{C}_l$ according to Lemma B.5:*

$$\frac{\pi^{\frac{d_l}{2}}}{\Gamma\left(\frac{d_l}{2}\right)}\left(\frac{1}{2^l\sigma^2}\right)^{-\frac{d_l}{2}}\prod_{i=1}^{l}\Gamma\left(\frac{d_i}{2}\right). \tag{92}$$

*proof of lemma B.4.* The normalization constant has the following form:

$$C = \int_{\mathbb{R}^{d_l}} \tilde{p}(\boldsymbol{f}^{(l)})d\boldsymbol{f}^{(l)} = \int_{\mathbb{R}^{d_l}} G_{0,l}^{l,0}\left(\frac{\|\boldsymbol{f}^{(l)}\|^2}{2^l\sigma^2}\bigg|0, \frac{1}{2}(d_1 - d_l)\ldots, \frac{1}{2}(d_{l-1} - d_l)\right)d\boldsymbol{f}^{(l)} \tag{93}$$

$$= \tilde{C}_l \int_0^\infty r^{d_l - 1} G_{0,l}^{l,0}\left(\frac{r^2}{2^l\sigma^2}\bigg|0, \frac{1}{2}(d_1 - d_l)\ldots, \frac{1}{2}(d_{l-1} - d_l)\right)dr \tag{94}$$

$$= \frac{1}{2}\tilde{C}_l \int_0^\infty x^{\frac{d_l}{2} - 1} G_{0,l}^{l,0}\left(\frac{x}{2^l\sigma^2}\bigg|0, \frac{1}{2}(d_1 - d_l)\ldots, \frac{1}{2}(d_{l-1} - d_l)\right)dx \tag{95}$$

$$= \frac{1}{2}\tilde{C}_l\left(\frac{1}{2^l\sigma^2}\right)^{-\frac{d_l}{2}}\prod_{i=1}^{l}\Gamma\left(\frac{d_i}{2}\right), \tag{96}$$

where we used spherical coordinates and the substitution $x = r^2$. We denote the angular constant by $\tilde{C}_l$, and according to Lemma B.5 has solution: $\tilde{C}_l = \frac{2\pi^{\frac{d_l}{2}}}{\Gamma\left(\frac{d_l}{2}\right)}$.

$\qquad\square$

## B.7 Angular constant

**Lemma B.5.** *The angular constant:*

$$\tilde{C}_l = \int_0^{2\pi} d\gamma_{d_l-1} \int_0^\pi \sin^{d_l-2}(\gamma_1)d\gamma_1 \int_0^\pi \sin^{d_l-3}(\gamma_2)d\gamma_2 \cdots \int_0^\pi \sin(\gamma_{d_l-2})d\gamma_{d_l-2} \tag{97}$$

*has solution:*

$$\tilde{C}_l = \frac{2\pi^{\frac{d_l}{2}}}{\Gamma\left(\frac{d_l}{2}\right)}. \tag{98}$$

*Proof.* The angular constant $\tilde{C}_l$ can be calculated as follows (for $d_l \geq 2$):

$$\tilde{C}_l = \int_0^{2\pi} d\gamma_{d_l-1} \int_0^\pi \sin^{d_l-2}(\gamma_1)d\gamma_1 \int_0^\pi \sin^{d_l-3}(\gamma_2)d\gamma_2 \cdots \int_0^\pi \sin(\gamma_{d_l-2})d\gamma_{d_l-2}$$

$$= 2\pi \prod_{k=1}^{d_l-2} \int_0^\pi \sin^{d_l-k-1}(\gamma_k)d\gamma_k$$

$$= 2\pi \prod_{k=1}^{d_l-2} \frac{\Gamma(d_l-k-1)}{2^{d_l-k-1}\Gamma\left(\frac{d_l-k-1}{2}\right)\Gamma\left(\frac{d_l-k-1}{2}+1\right)}(2\pi)$$

$$= \frac{(2\pi)^{d_l-1}}{2^{\frac{1}{2}(d_l-2)(d_l-1)}} \prod_{k=1}^{d_l-2} \frac{\Gamma(d_l-k-1)}{\Gamma\left(\frac{d_l-k-1}{2}\right)\Gamma\left(\frac{d_l-k-1}{2}+1\right)},$$

where we have used Lemma B.6 to compute the integrals. If $d_l = 1$, then there is no need to write the integral in spherical coordinates and we can simply set $\tilde{C} = 1$.

Now we can apply the Legendre duplication formula:

$$\Gamma(z)\Gamma(z+\frac{1}{2}) = 2^{1-2z}\sqrt{\pi}\Gamma(2z) \tag{99}$$

to the numerator term $2z = d_l - k - 1$ and get:

$$\tilde{C}_l = \frac{(2\pi)^{d_l-1}}{\pi^{\frac{d_l-2}{2}}2^{\frac{1}{2}(d_l-2)(d_l-1)}} \prod_{k=1}^{d_l-2} \frac{\Gamma\left(\frac{d_l-k-1}{2}\right)\Gamma(\frac{d_l-k-1}{2}+\frac{1}{2})2^{d_l-k-2}}{\Gamma\left(\frac{d_l-k-1}{2}\right)\Gamma\left(\frac{d_l-k-1}{2}+1\right)}. \tag{100}$$

Note that that the product:

$$\prod_{k=1}^{d_l-2} \frac{\Gamma(\frac{d_l-k-1}{2}+\frac{1}{2})}{\Gamma\left(\frac{d_l-k-1}{2}+1\right)} = \frac{\Gamma(\frac{d_l-1}{2})}{\Gamma(\frac{d_l}{2})} \cdot \frac{\Gamma(\frac{d_l-2}{2})}{\Gamma(\frac{d_l-1}{2})} \cdots \frac{1}{\Gamma(\frac{1}{2})} = \frac{1}{\Gamma(\frac{d_l}{2})} \tag{101}$$

Finally, we have the product

$$\prod_{k=1}^{d_l-2} 2^{d_l-k-2} = 2^{d_l(d_l-2)-\frac{(d_l-2)(d_l-1)}{2}-2(d_l-2)}, \tag{102}$$

from which we can conclude, after some elementary algebraic manipulations:

$$\tilde{C}_l = \frac{2\pi^{\frac{d_l}{2}}}{\Gamma\left(\frac{d_l}{2}\right)}. \tag{103}$$

$\square$

Finally we prove the technical Lemma that we used in the previous proof.

**Lemma B.6.**

$$\int_0^\pi \sin^k(x)dx = \frac{\Gamma(k)}{2^k\Gamma\left(\frac{k}{2}\right)\Gamma\left(\frac{k}{2}+1\right)}(2\pi). \tag{104}$$

*Proof.* By integrating by parts, and using some algebraic manipulation, it is easy to see that:

$$\int \sin^k(x)dx = -\frac{1}{k}\sin^{k-1}x\cos x + \frac{k-1}{k}\int \sin^{k-2}xdx. \tag{105}$$

Evaluating the integral between $0$ and $\pi$, we get:

$$\int_0^\pi \sin^k(x)dx = \frac{k-1}{k}\int_0^\pi \sin^{k-2}xdx. \tag{106}$$

By unrolling the recursion:

$$\int_0^\pi \sin^k(x)dx = \frac{(k-1)(k-3)\cdots}{k(k-2)\cdots}\begin{cases} \int_0^\pi dx = \pi & \text{if } k \text{ is even} \\ \int_0^\pi \sin(x)dx = 2 & \text{if } k \text{ is odd} \end{cases}. \tag{107}$$

The following expression includes both the even and the odd case:

$$\int_0^\pi \sin^k(x)dx = \frac{\Gamma(k)}{2^k\Gamma\left(\frac{k}{2}\right)\Gamma\left(\frac{k}{2}+1\right)}(2\pi) \tag{108}$$

In fact, if $k$ is even, then:

$$\frac{\Gamma(k)}{2^k\Gamma\left(\frac{k}{2}\right)\Gamma\left(\frac{k}{2}+1\right)}(2\pi) = \frac{(k-1)(k-2)\cdots}{2^k\frac{k}{2}(\frac{k}{2}-1)^2(\frac{k}{2}-2)^2\cdots}(2\pi)$$

$$= \frac{(k-1)(k-3)\cdots}{k(k-2)\cdots}\pi$$

If $k$ is odd, we use the identity:

$$\Gamma\left(\frac{k}{2}\right) = \frac{(k-2)!!\sqrt{\pi}}{2^{\frac{k-1}{2}}}, \tag{109}$$

where $k!!$ is the double factorial. Following a very similar procedure, we get the desired result. $\square$

**Important Remark:** In the ReLU case, we will see that that the integral is from $0$ to $\frac{\pi}{2}$. In that case, we get:

$$\int_0^{\frac{\pi}{2}} \sin^k(x)dx = \frac{\Gamma(k)}{2^k\Gamma\left(\frac{k}{2}\right)\Gamma\left(\frac{k}{2}+1\right)}\pi. \tag{110}$$

Therefore the angular constant is:

$$\tilde{C}_l = \int_0^{\frac{\pi}{2}}d\gamma_{d_l-1}\int_0^{\frac{\pi}{2}}\sin^{d_l-2}(\gamma_1)d\gamma_1\int_0^{\frac{\pi}{2}}\sin^{d_l-3}(\gamma_2)d\gamma_2\cdots\int_0^{\frac{\pi}{2}}\sin(\gamma_{d_l-2})d\gamma_{d_l-2} \tag{111}$$

$$= \frac{\pi}{2}\prod_{k=1}^{d_l-2}\int_0^{\frac{\pi}{2}}\sin^{d_l-k-1}(\gamma_k)d\gamma_k \tag{112}$$

$$= \frac{\pi^{d_l-1}}{2}\prod_{k=1}^{d_l-2}\frac{1}{2^{d_l-k-1}}\frac{\Gamma(d_l-k-1)}{\Gamma\left(\frac{d_l-k-1}{2}\right)\Gamma\left(\frac{d_l-k-1}{2}+1\right)} \tag{113}$$

$$= \frac{\pi^{d_l-1}}{2\pi^{\frac{d_l-2}{2}}}\prod_{k=1}^{d_l-2}\frac{\Gamma(\frac{d_l-k-1}{2}+\frac{1}{2})2^{-1}}{\Gamma\left(\frac{d_l-k-1}{2}+1\right)} \tag{114}$$

$$= \frac{\pi^{d_l-1}}{2\pi^{\frac{d_l-2}{2}}}\frac{2^{-(d_l-2)}}{\Gamma(\frac{d_l}{2})} \tag{115}$$

$$= \frac{\pi^{\frac{d_l}{2}}}{2^{d_l-1}\Gamma(\frac{d_l}{2})}. \tag{116}$$

## B.8 Kurtosis of Linear Networks

Using the closed form expressions from Section B.4, we can describe the kurtosis of the output $f^{(l)}$ as

$$\kappa_{\text{lin}} := \frac{3\sigma_w^{2l}(m+2)^{l-1}}{\sigma_w^{2l}m^{l-1}} = 3\left(\frac{m+2}{m}\right)^{l-1}. \tag{117}$$

In particular, the distribution is always more heavy-tailed than a Gaussian, for which $\kappa = 3$ (i.e. the distribution is leptokurtic). The second obvious conclusion is that depth increases the heavy-tailedness exponentially, which is in-line with the theoretical results of (Vladimirova et al., 2019). On the contrary, the width has the effect of "normalizing" the distribution, in particular in the limit of large width we have that:

$$\lim_{m\to\infty}\kappa = 3, \tag{118}$$

which is the kurtosis of the Gaussian distribution, as anticipated from Lemma B.3.

# C Proofs for ReLU networks and Derivation of their Moments

Now, we extend these results to ReLU networks. We need the following additional notation: we call $\delta(x - x_0)$ the Dirac delta function centered at $x_0$, and $\mathbb{1}_A := \begin{cases} 1 & x \in A \\ 0 & \text{else} \end{cases}$ the indicator function. Also, we indicate with $\mathcal{S}$ the set of indices $1, \ldots, d$ that index $d$ random variables, and with $\Omega$ its power set, i.e. the set of all possible subsets of $\mathcal{S}$. Note that $|\Omega| = 2^d$. We will use the following lemma, which explains what happens to a joint density when the marginals are transformed by the ReLU function.

## C.1 Effect of ReLU activation function on the joint density

**Lemma C.1.** *Let $p(f_1, \ldots f_d)$ be the joint density of the $d$ random variables $f_1, \ldots f_d$.[a] Assume $p(f_1, \ldots f_d)$ is symmetric around zero. When we apply the transformation $g_i = ReLU(f_i)$ to each $i \in [d]$, then the joint density of the transformed variables has the following form:*

$$p_{ReLU}(g_1, \ldots g_d) = \sum_{A \in \Omega} \frac{1}{2^{|A|}} p(\boldsymbol{g}_{\mathcal{S}\backslash A}) \prod_{j \in \mathcal{S}\backslash A} \mathbb{1}_{g_j > 0} \prod_{i \in A} \delta(g_i), \tag{119}$$

*where $p(\boldsymbol{g}_{\mathcal{S}\backslash A})$ is the marginal density of the random variables whose indexes are in $\mathcal{S} \setminus A$.*

---

[a]We use the same notation for the random variable and the corresponding dummy variable in the density function.

*Proof.* Again we use conditional independence: the activations $g_1, \ldots, g_d$ are independent given the pre-activations $f_1, \ldots, f_d$. So we can write:

$$p_{\text{ReLU}}(g_1, \ldots, g_d) = \int_{\mathbb{R}^d} p(g_1, \ldots, g_d | f_1, \ldots, f_d) p(f_1, \ldots, f_d) d\boldsymbol{f} \tag{120}$$

$$= \int_{\mathbb{R}^d} \prod_{i=1}^d p(g_i | f_i) p(f_1, \ldots, f_d) d\boldsymbol{f}. \tag{121}$$

Now, $p(g_i | f_i) = \begin{cases} \delta(g_i) & f_i < 0 \\ \delta(g_i - f_i) & f_i \geq 0 \end{cases} = \delta(g_i)\mathbb{1}_{f_i < 0} + \delta(g_i - f_i)\mathbb{1}_{f_i \geq 0}$. So we can write:

$$p_{\text{ReLU}}(g_1, \ldots, g_d) = \int_{\mathbb{R}^d} \prod_{i=1}^{d} \left( \delta(g_i) \mathbb{1}_{f_i < 0} + \delta(g_i - f_i) \mathbb{1}_{f_i \geq 0} \right) p(f_1, \ldots, f_d) d\boldsymbol{f} \tag{122}$$

$$= \int_{\mathbb{R}^d} \prod_{i=1}^{d} \left( \delta(g_i) \mathbb{1}_{f_i < 0} + \delta(g_i - f_i) \mathbb{1}_{f_i \geq 0} \right) p(f_1, \ldots, f_d) d\boldsymbol{f} \tag{123}$$

$$= \int_{\mathbb{R}^d} \sum_{A \in \Omega} \left( \prod_{i \in A} \delta(g_i) \mathbb{1}_{f_i < 0} \prod_{j \in \mathcal{S} \setminus A} \delta(g_j - f_j) \mathbb{1}_{f_j \geq 0} \right) p(f_1, \ldots, f_d) d\boldsymbol{f} \tag{124}$$

$$= \sum_{A \in \Omega} \int_{\mathbb{R}_{<0}^{|A|}} \int_{\mathbb{R}^{d-|A|}} \prod_{i \in A} \delta(g_i) \prod_{j \in \mathcal{S} \setminus A} \delta(g_j - f_j) \mathbb{1}_{f_j \geq 0} p(f_1, \ldots, f_d) d\boldsymbol{f} \tag{125}$$

$$= \sum_{A \in \Omega} \prod_{i \in A} \delta(g_i) \int_{\mathbb{R}_{<0}^{|A|}} \int_{\mathbb{R}^{d-|A|}} \prod_{j \in \mathcal{S} \setminus A} \delta(g_j - f_j) \mathbb{1}_{f_j \geq 0} p(f_1, \ldots, f_d) d\boldsymbol{f} \tag{126}$$

$$= \sum_{A \in \Omega} \prod_{i \in A} \delta(g_i) \int_{\mathbb{R}_{<0}^{|A|}} p(\boldsymbol{f}_A, \boldsymbol{g}_{\mathcal{S} \setminus A}) d\boldsymbol{f}_A \prod_{j \in \mathcal{S} \setminus A} \mathbb{1}_{g_j > 0} \tag{127}$$

$$= \sum_{A \in \Omega} \frac{1}{2^{|A|}} \prod_{i \in A} \delta(g_i) p(\boldsymbol{g}_{\mathcal{S} \setminus A}) \prod_{j \in \mathcal{S} \setminus A} \mathbb{1}_{g_j > 0}, \tag{128}$$

where in the second to last step we have used the well known property of the Delta function $\int_{-\infty}^{\infty} f(x) \delta(x - x_0) dx = f(x_0)$ and in the last step we used the fact that the density $p$ is symmetric around 0. □

## C.2 Proof of Theorem 4.3

**Theorem.** *Suppose $l \geq 2$, and the input has dimension $d_0$. Define the multi-index set $\mathcal{R} = [d_1] \times \cdots \times [d_{l-1}]$ and introduce the vector $\boldsymbol{u^r} \in \mathbb{R}^{l-1}$ through its components $\boldsymbol{u}_i^{\boldsymbol{r}} = \frac{1}{2}(r_i - d_l)$.*

$$p(\boldsymbol{f}_{ReLU}^{(l)}) = \sum_{\boldsymbol{r} \in \mathcal{R}} q_{\boldsymbol{r}} G_{0,l}^{l,0} \left( \frac{\|\boldsymbol{f}_{ReLU}^{(l)}\|^2}{2^l \sigma^2} \middle| 0, \boldsymbol{u^r} \right) + q_0 \delta(\boldsymbol{f}_{ReLU}^{(l)}), \tag{129}$$

*where $\sigma^2 = \prod_{i=1}^{l} \sigma_i^2$ and the individual weights are given by*

$$q_{\boldsymbol{r}} = \pi^{-\frac{d_l}{2}} 2^{-\frac{l}{2} d_l} (\sigma^2)^{-\frac{d_l}{2}} \prod_{i=1}^{l-1} \binom{d_i}{r_i} \frac{1}{2^{d_i} \Gamma\left(\frac{r_i}{2}\right)}, \tag{130}$$

*and*

$$q_0 = 1 - \prod_{i=1}^{l-1} \frac{2^{d_i} - 1}{2^{d_i}} \tag{131}$$

*Proof.* Before starting the proof, note that there is a special case the has to be handled separately: the case in which *all units* are inactive, i.e., the ReLU activation sets to zero all the pre-activation in a layer. This will be handled at end of the proof. First, let's assume that there is at least one active unit per layer.

The proof is again by induction. The base case ($l = 2$) is stated in Lemma C.2. For the general case, we use again an identical approach as in Theorem 4.1. We expand the coefficients and write $c_{r_l}^l := \binom{d_l}{r_l} \frac{1}{2^{d_l} \Gamma\left(\frac{d_l - r_l}{2}\right)}$, where $l > 0$ is the layer index. Induction step: assume that the

pre-nonlinearities have the following form:

$$p(f_1^{(l-1)}, \ldots, f_{d_{l-1}}^{(l-1)}) = \sum_{r_1=0}^{d_1-1} c_{r_1}^1 \cdots \sum_{r_{l-2}=0}^{d_{l-2}-1} c_{r_{l-2}}^{l-2} \pi^{-\frac{d_{l-1}}{2}} 2^{-\frac{(l-1)d_{l-1}}{2}} (\tilde{\sigma}^2)^{-\frac{d_{l-1}}{2}} \tag{132}$$

$$G_{0,l-1}^{l-1,0}\left(\frac{||\boldsymbol{f}^{(l-1)}||^2}{2^{l-1}\tilde{\sigma}^2}\Big|0, \frac{1}{2}(d_1 - r_1 - d_{l-1}), \ldots, \frac{1}{2}(d_{l-2} - r_{l-2} - d_{l-1})\right), \tag{133}$$

where $\tilde{\sigma}^2 = \prod_{i=1}^{l-1} \sigma_i^2$. We know from Lemma C.1, that the activations $g_1^{(l-1)}, \ldots, g_{d_{l-1}}^{(l-1)}$ have the following density:

$$p_{\text{ReLU}}(g_1^{(l-1)}, \ldots, g_{d_{l-1}}^{(l-1)}) = \sum_{A \in \Omega} \frac{1}{2^{|A|}} \prod_{i \in A} \delta(g_i^{(l-1)}) p(\boldsymbol{g}_{\mathcal{S}\backslash A}^{(l-1)}) \prod_{j \in \mathcal{S}\backslash A} \mathbb{1}_{g_j > 0} \tag{134}$$

$$= \sum_{A \in \Omega} \frac{1}{2^{|A|}} \sum_{r_1=0}^{d_1-1} c_{r_1}^1 \cdots \sum_{r_{l-2}=0}^{d_{l-2}-1} c_{r_{l-2}}^{l-2} \pi^{-\frac{d_{l-1}}{2}} 2^{-\frac{(l-1)d_{l-1}}{2}} (\tilde{\sigma}^2)^{-\frac{d_{l-1}}{2}} \tag{135}$$

$$G_{0,l-1}^{l-1,0}\left(\frac{||\boldsymbol{g}_{\mathcal{S}\backslash A}^{(l-1)}||^2}{2^{l-1}\tilde{\sigma}^2}\Big|0, \frac{1}{2}(d_1 - r_1 - d_A), \ldots, \frac{1}{2}(d_{l-2} - r_{l-2} - d_A)\right) \tag{136}$$

$$\prod_{i \in A} \delta(g_i^{(l-1)}) \prod_{j \in \mathcal{S}\backslash A} \mathbb{1}_{g_j > 0}, \tag{137}$$

where $d_A := |\mathcal{S} \setminus A| = d_{l-1} - |A|$. Also, here we abuse the notation and consider that $\mathcal{S}$ is not in the power set, i.e., $\mathcal{S} \notin \Omega$. This is to be consistent with the fact that we are handling the case in which at least one unit is active after the ReLU activation is applied. Now following a similar procedure as in Lemma C.2, we have

$$p(f_1^{(l)}, \ldots f_{d_l}^{(l)}) = \int_{\mathbb{R}_{\geq 0}^{d_{l-1}}} p(f_1^{(l)}, \ldots f_{d_l}^{(l)}|g_1^{(l-1)}, \ldots g_{d_1}^{(l-1)}) p_{\text{ReLU}}(g_1^{(l-1)}, \ldots g_{d_1}^{(l-1)}) d\boldsymbol{g}^{(l-1)}, \tag{138}$$

which is equal to:

$$\int_{\mathbb{R}_{\geq 0}^{d_{l-1}}} \frac{1}{(2\pi\sigma_l^2 ||\boldsymbol{g}^{(l-1)}||^2)^{\frac{d_l}{2}}} e^{-\frac{||\boldsymbol{f}^{(l)}||^2}{2\sigma_l^2 ||\boldsymbol{g}^{(l-1)}||^2}} \sum_{A \in \Omega} \frac{1}{2^{|A|}} \sum_{r_1=0}^{d_1-1} c_{r_1}^1 \cdots \sum_{r_{l-2}=0}^{d_{l-2}-1} c_{r_{l-2}}^{l-2} \tag{139}$$

$$\pi^{-\frac{d_{l-1}-|A|}{2}} 2^{-\frac{(l-1)(d_{l-1}-|A|)}{2}} (\tilde{\sigma}^2)^{-\frac{d_{l-1}-|A|}{2}} \tag{140}$$

$$G_{0,l-1}^{l-1,0}\left(\frac{||\boldsymbol{g}_{\mathcal{S}\backslash A}^{(l-1)}||^2}{2^{l-1}\tilde{\sigma}^2}\Big|0, \frac{1}{2}(d_1 - r_1 - d_A), \ldots, \frac{1}{2}(d_{l-2} - r_{l-2} - d_A)\right) \tag{141}$$

$$\prod_{i \in A} \delta(g_i^{(l-1)}) \prod_{j \in \mathcal{S}\backslash A} \mathbb{1}_{g_j > 0} d\boldsymbol{g}^{(l-1)} \tag{142}$$

$$= \sum_{A \in \Omega} \frac{\pi^{-\frac{d_{l-1}-|A|}{2}} 2^{-\frac{(l-1)(d_{l-1}-|A|)}{2}} (\tilde{\sigma}^2)^{-\frac{d_{l-1}-|A|}{2}}}{2^{|A|}(2\pi\sigma_l^2)^{\frac{d_l}{2}}} \sum_{r_1=0}^{d_1-1} c_{r_1}^1 \cdots \sum_{r_{l-2}=0}^{d_{l-2}-1} c_{r_{l-2}}^{l-2} \tag{143}$$

$$\int_{\mathbb{R}_{>0}^{d_{l-1}}} \frac{1}{(||\boldsymbol{g}_{\mathcal{S}\backslash A}^{(l-1)}||^2)^{\frac{d_l}{2}}} e^{-\frac{||\boldsymbol{f}^{(l)}||^2}{2\sigma_l^2 ||\boldsymbol{g}_{\mathcal{S}\backslash A}^{(l-1)}||^2}} \tag{144}$$

$$G_{0,l-1}^{l-1,0}\left(\frac{||\boldsymbol{g}_{\mathcal{S}\backslash A}^{(l-1)}||^2}{2^{l-1}\tilde{\sigma}^2}\Big|0, \frac{1}{2}(d_1 - r_1 - d_A), \ldots, \frac{1}{2}(d_{l-2} - r_{l-2} - d_A)\right) d\boldsymbol{g}_{\mathcal{S}\backslash A}^{(l-1)} \tag{145}$$

For each set $A$ we have a $(d_{l-1} - |A|)$- dimensional integral that can be solved using once again Lemma 4.2 [2]. Note that for the new Meijer-G coefficients of Lemma 4.2:

$$\frac{1}{2}(d_i - r_i - d_{l-1} + |A|) + \frac{1}{2}(d_{l-1} - |A| - d_l) = \frac{1}{2}(d_i - r_i - d_l) \tag{146}$$

holds for all $i \in [d_{l-2}]$. Therefore the solution of each integral is equal to

$$\frac{1}{2}\tilde{C}_A 2^{\frac{1}{2}(d_{l-1} - |A| - d_l)(l-1)} \tilde{\sigma}^{(d_{l-1} - |A| - d_l)} \tag{147}$$

$$G_{0,l}^{l,0}\left(\frac{||\boldsymbol{f}^{(l)}||^2}{2^l \tilde{\sigma}^2}\bigg| 0, \frac{1}{2}(d_1 - r_1 - d_l), \ldots, \frac{1}{2}(d_{l-1} - r_{l-1} - d_l)\right). \tag{148}$$

The new coefficient for every set $A$ is:

$$c_A = \frac{\pi^{-\frac{d_{l-1} - |A|}{2}} 2^{-\frac{(l-1)(d_{l-1} - |A|)}{2}} (\tilde{\sigma}^2)^{-\frac{d_{l-1} - |A|}{2}}}{2^{|A|}(2\pi\sigma_l^2)^{\frac{d_l}{2}}} \frac{1}{2}\tilde{C}_A 2^{\frac{1}{2}(d_{l-1} - |A| - d_l)(l-1)} \tilde{\sigma}^{(d_{l-1} - |A| - d_l)} \tag{149}$$

$$= \frac{\cancel{\pi^{-\frac{d_{l-1} - |A|}{2}}} 2^{-\frac{(l-1)(d_{l-1} - |A|)}{2}} \cancel{(\tilde{\sigma}^2)^{-\frac{d_{l-1} - |A|}{2}}}}{2^{\cancel{|A|}}(2\pi\sigma_l^2)^{\frac{d_l}{2}}} \frac{1}{\cancel{2}} \frac{\cancel{\pi^{-\frac{d_{l-1} - |A|}{2}}}}{2^{d_{l-1} - \cancel{|A|} - \cancel{1}}\Gamma\left(\frac{d_{l-1} - |A|}{2}\right)} \tag{150}$$

$$\tilde{C}_A 2^{\frac{1}{2}(\cancel{d_l} + \cancel{|A|} - d_l)(l-1)} \tilde{\sigma}^{(\cancel{d_l} + \cancel{|A|} - d_l)} \tag{151}$$

$$= \frac{\pi^{-\frac{d_l}{2}} 2^{-\frac{ld_l}{2}}(\sigma^2)^{-\frac{d_l}{2}}}{2^{d_{l-1}}\Gamma\left(\frac{d_{l-1} - |A|}{2}\right)} \tag{152}$$

Therefore, because the dependence on $A$ is only through its cardinality $r_{l-1}^{l-1} := |A|$, we define:

$$c_{r_{l-1}}^{l-1} := \binom{d_{l-1}}{r_{l-1}} \frac{1}{2^{d_{l-1}}\Gamma\left(\frac{d_{l-1} - r_{l-1}}{2}\right)} \tag{153}$$

So the solution is:

$$p(f_1^{(l)}, \ldots, f_{d_l}^{(l)}) = \sum_{r_1=0}^{d_1-1} c_{r_1}^1 \cdots \sum_{r_{l-1}=0}^{d_{l-1}-1} c_{r_{l-1}}^{l-1} \pi^{-\frac{d_l}{2}} 2^{-\frac{ld_l}{2}}(\sigma^2)^{-\frac{d_l}{2}} \tag{154}$$

$$G_{0,l}^{l,0}\left(\frac{||\boldsymbol{f}^{(l)}||^2}{2^l \sigma^l}\bigg| 0, \frac{1}{2}(d_1 - r_1 - d_l), \ldots, \frac{1}{2}(d_{l-1} - r_{l-1} - d_l)\right). \tag{155}$$

The final form of this equation stated in the theorem is obtained by grouping all the coefficients not involving the Meijer-G function, and substituting $r_i \leftarrow d_i - r_i$ and use the property $\binom{d_i}{r_i} = \binom{d_i}{d_i - r_i}$.

**Special case: all units are inactive** If at least in one layer it happens that all post-activations are zero, then the distribution of the network is a point mass at 0. Let's call this event $E$, and its probability $q_0$. The probability of its complement $\bar{E}$ is the probability that for all the intermediate layers, at least one unit is active. These are $l - 1$ independent events, the probability of each being $\frac{2^{d_i} - 1}{2^{d_i}}$ (one unit is active in $2^{d_i} - 1$ cases out of the all possible combinations of units). Therefore we can conclude that:

$$q_0 = 1 - \prod_{i=1}^{l-1} \frac{2^{d_i} - 1}{2^{d_i}} \tag{156}$$

$\square$

---

[2] see proof of Lemma C.2 for a small but important detail of this integral

## C.3 Base case for ReLU nets

**Lemma C.2** (second layer pre-activations density). *Let $\sigma = \sigma_1\sigma_2$. Conditioned on the event that at least one unit of the first layer is active ($g_j^{(1)} \neq 0$ for at least one $j \in [d_1]$), the density of the second layer's pre-activations $f_1^{(2)}, \ldots f_{d_2}^{(2)}$ is the following linear combination of Meijer-G functions:*

$$p(\boldsymbol{f}_1^{(2)}) = \sum_{r=0}^{d_1-1} c_r 2^{-d_2} (\sigma^2)^{-\frac{d_2}{2}} \pi^{\frac{-d_2}{2}} G_{0,2}^{2,0} \left( \frac{||\boldsymbol{f}^{(2)}||^2}{4\sigma^2} \middle| 0, \frac{1}{2}((d_1-r)-d_2) \right) + q_0 \delta(\boldsymbol{f}_1^{(2)}), \quad (157)$$

*where $c_r := \binom{d_1}{r} \frac{1}{2^{d_1}\Gamma\left(\frac{d_1-r}{2}\right)}$ and $q_0 := 1 - \frac{2^{d_1-1}}{2^{d_i}}$.*

*Proof.*

$$p(f_1^{(2)}, \ldots f_{d_2}^{(2)}) = \int_{\mathbb{R}_{\geq 0}^{d_1}} p(f_1^{(2)}, \ldots f_{d_2}^{(2)}|g_1^{(1)}, \ldots g_{d_1}^{(1)}) p_{\text{ReLU}}\left(g_1^{(1)}, \ldots g_{d_1}^{(1)}\right) d\boldsymbol{g}^{(1)} \quad (158)$$

$$= \int_{\mathbb{R}_{\geq 0}^{d_1}} \prod_{k=1}^{d_2} p(f_k^{(2)}|\boldsymbol{g}^{(1)}) \prod_{k'=1}^{d_1} p_{ReLU}\left(g_{k'}^{(1)}\right) d\boldsymbol{g}^{(1)} \quad (159)$$

$$= \int_{\mathbb{R}_{\geq 0}^{d_1}} \frac{1}{(2\pi\sigma_2^2||\boldsymbol{g}^{(1)}||^2)^{\frac{d_2}{2}}} \exp\left( -\frac{||\boldsymbol{f}^{(2)}||^2}{2\sigma_2^2||\boldsymbol{g}^{(1)}||^2} \right) \cdot \quad (160)$$

$$\sum_{A\in\Omega} \frac{1}{2^{|A|}} \prod_{i\in A} \delta(g_i^{(1)}) \frac{1}{(2\pi\sigma_1^2)^{\frac{d_1-|A|}{2}}} \exp\left( -\frac{\sum_{i\in\mathcal{S}\setminus A}(g_i^{(1)})^2}{2\sigma_1^2} \right) \prod_{i\in\mathcal{S}\setminus A} \mathbb{1}_{g_i^{(1)}>0} d\boldsymbol{g}^{(1)} \quad (161)$$

$$= \sum_{A\in\Omega} \frac{1}{2^{|A|}} \frac{1}{(2\pi\sigma_1^2)^{\frac{d_1-|A|}{2}}(2\sigma_2^2)^{\frac{d_2}{2}}} \int_{\mathbb{R}_{\geq 0}^{d_1}} \frac{1}{(||\boldsymbol{g}^{(1)}||^2)^{\frac{d_2}{2}}} \exp\left( -\frac{||\boldsymbol{f}^{(2)}||^2}{2\sigma_2^2||\boldsymbol{g}^{(1)}||^2} \right) \quad (162)$$

$$\prod_{i\in A} \delta(g_i^{(1)}) \exp\left( -\frac{\sum_{i\in\mathcal{S}\setminus A}(g_i^{(1)})^2}{2\sigma_1^2} \right) \prod_{i\in\mathcal{S}\setminus A} \mathbb{1}_{g_i^{(1)}>0} d\boldsymbol{g}^{(1)}, \quad (163)$$

where we can exchange sum and integration due to non-negativeness of the integration variables (Tonelli's theorem). Also, here we abuse the notation and consider that $\mathcal{S}$ is not in the power set, i.e., $\mathcal{S} \notin \Omega$. This is to be consistent with the fact that we are conditioning on the event in which at least one unit is active after the ReLU activation is applied in the first layer. Now we can use the property of the delta function $\int f(x)\delta(x-x_0)dx = f(x_0)$ and the property of the indicator function $\int_A f(x)\mathbb{1}_{x\in B}dx = \int_B f(x)dx$ and get:

$$\sum_{A\in\Omega} \frac{1}{2^{|A|}} \frac{1}{(2\pi\sigma_1^2)^{\frac{d_1-|A|}{2}}(2\sigma_2^2)^{\frac{d_2}{2}}} \quad (164)$$

$$\int_{\mathbb{R}_{>0}^{d_1}} \frac{1}{(||\boldsymbol{g}_{\mathcal{S}\setminus A}^{(1)}||^2)^{\frac{d_2}{2}}} \exp\left( -\frac{||\boldsymbol{f}^{(2)}||^2}{2\sigma_2^2||\boldsymbol{g}_{\mathcal{S}\setminus A}^{(1)}||^2} \right) \exp\left( -\frac{||\boldsymbol{g}_{\mathcal{S}\setminus A}^{(1)}||^2}{2\sigma_1^2} \right) d\boldsymbol{g}_{\mathcal{S}\setminus A}^{(1)}. \quad (165)$$

Note that the above integral is $(d_1-|A|)$ dimensional due to the effect of the delta. Now the integral(s) above can be solved in an equivalent manner as in the previous section using Lemma 4.2[3], and they are equal to

$$\frac{1}{2}\tilde{C}_A (2\sigma_1^2)^{\frac{1}{2}(d_1-|A|-d_2)} G_{0,2}^{2,0}\left( \frac{||\boldsymbol{f}^{(2)}||^2}{4\sigma^2} \middle| 0, \frac{1}{2}((d_1-|A|)-d_2) \right) \quad (166)$$

---

[3]Note that the integral is only for the positive reals. Lemma 4.2 can still be used because when switching to spherical coordinates, we are interested in the radius part, while the angular constant can still be calculated, but now we the angles are all from 0 to $\frac{\pi}{2}$

So we can conclude that:

$$p(f_1^{(2)}, \dots f_{d_2}^{(2)}) = \sum_{A \in \Omega} \frac{1}{2^{|A|}} \frac{1}{2} \tilde{C}_A (2\sigma_1^2)^{\frac{1}{2}(d_1 - |A| - d_2)} \frac{1}{(2\pi\sigma_1^2)^{\frac{d_1 - |A|}{2}} (2\sigma_2^2)^{\frac{d_2}{2}}} \tag{167}$$

$$G_{0,2}^{2,0} \left( \frac{||\boldsymbol{f}^{(2)}||^2}{4\sigma^2} \bigg| 0, \frac{1}{2}((d_1 - |A|) - d_2) \right) \tag{168}$$

$$= \pi^{-\frac{d_1}{2}} \sum_{r=0}^{d_1-1} \binom{d_1}{r} \frac{\tilde{C}_r}{2^{r+1}(2\sigma_1^2)^{\frac{d_2}{2}} (2\sigma_2^2)^{\frac{d_2}{2}} \pi^{\frac{-r+d_2}{2}}} \tag{169}$$

$$G_{0,2}^{2,0} \left( \frac{||\boldsymbol{f}^{(2)}||^2}{4\sigma^2} \bigg| 0, \frac{1}{2}((d_1 - r) - d_2) \right), \tag{170}$$

where we have used the fact that the expression depends on the set $A \in \Omega$ only through $|A|$, and therefore we can use the fact that the number of subsets with $r$ elements is given by the binomial coefficient $\binom{d_1}{r}$. Define:

$$c_r := \binom{d_1}{r} \pi^{-\frac{d_1}{2}} \frac{\tilde{C}_r}{2^{r+1}} \pi^{\frac{r}{2}} \tag{171}$$

$$= \binom{d_1}{r} \pi^{-\frac{d_1}{2}} \frac{\pi^{\frac{d_1-r}{2}}}{2^{d_1-r-1}\Gamma\left(\frac{d_1-r}{2}\right)} \frac{1}{2^{r+1}} \pi^{\frac{r}{2}} \tag{172}$$

$$= \binom{d_1}{r} \frac{1}{2^{d_1}\Gamma\left(\frac{d_1-r}{2}\right)}. \tag{173}$$

So we can conclude:

$$p(f_1^{(2)}, \dots f_{d_2}^{(2)}) = \sum_{r=0}^{d_1-1} c_r 2^{-d_2} (\sigma^2)^{\frac{-d_2}{2}} \pi^{\frac{-d_2}{2}} G_{0,2}^{2,0} \left( \frac{||\boldsymbol{f}^{(2)}||^2}{4\sigma^2} \bigg| 0, \frac{1}{2}((d_1 - r) - d_2) \right). \tag{174}$$

Finally, there is the special case where all the units are inactive (set to zero). This happens with probability $q_0 = \frac{1 - 2^{d_i - 1}}{2^{d_i}}$. $\qquad\square$

**Remark** Any non empty subset of $d < d_2$ units has the same distribution (with terms involving $d_2$ replaced by $d$).

## C.4  Resulting moments

Let $d_l = 1$, $d_1, \dots, d_{l-1} = m$, and $b_i = \frac{1}{2}(r_i - 1)$, $i = 1, \dots, l - 1$.

$$\mathbb{E}\left[Z^{2k}\right] = \int_{\mathbb{R}} z^{2k} \sum_{r_1=1}^{m} c_{r_1}^1 \cdots \sum_{r_{l-1}=1}^{m} c_{r_{l-1}}^{l-1} \pi^{-\frac{1}{2}} 2^{\frac{l}{2}} (\sigma^2)^{-\frac{1}{2}} G_{0,l}^{l,0} \left( \frac{z^2}{2^l \sigma^2} \bigg| 0, b_1, \dots, b_{l-1} \right) dz$$

$$= \sum_{r_1=1}^{m} c_{r_1}^1 \cdots \sum_{r_{l-1}=1}^{m} c_{r_{l-1}}^{l-1} \pi^{-\frac{1}{2}} 2^{\frac{l}{2}} (\sigma^2)^{-\frac{1}{2}} \int_0^{\infty} r^{2k} G_{0,l}^{l,0} \left( \frac{r^2}{2^l \sigma^2} \bigg| 0, b_1, \dots, b_{l-1} \right) dr$$

$$= \sum_{r_1=1}^{m} c_{r_1}^1 \cdots \sum_{r_{l-1}=1}^{m} c_{r_{l-1}}^{l-1} \pi^{-\frac{1}{2}} 2^{\frac{l}{2}} (\sigma^2)^{-\frac{1}{2}} \int_0^{\infty} x^{k-\frac{1}{2}} G_{0,l}^{l,0} \left( \frac{x}{2^l \sigma^2} \bigg| 0, b_1, \dots, b_{l-1} \right) dx$$

$$= \sum_{r_1=1}^{m} c_{r_1}^1 \cdots \sum_{r_{l-1}=1}^{m} c_{r_{l-1}}^{l-1} \pi^{-\frac{1}{2}} 2^{\frac{l}{2}} (\sigma^2)^{-\frac{1}{2}} \left( \frac{1}{2^l \sigma^2} \right)^{-k-\frac{1}{2}} \Gamma\left(k + \frac{1}{2}\right) \prod_{i=1}^{l-1} \Gamma\left(k + \frac{1}{2} + b_i\right)$$

$$= \pi^{-\frac{1}{2}} 2^{\frac{l}{2}} (\sigma^2)^{-\frac{1}{2}} \left( \frac{1}{2^l \sigma^2} \right)^{-k-\frac{1}{2}} \Gamma\left(k + \frac{1}{2}\right) \sum_{r_1=1}^{m} c_{r_1}^1 \cdots \sum_{r_{l-1}=1}^{m} c_{r_{l-1}}^{l-1} \prod_{i=1}^{l-1} \Gamma\left(k + \frac{1}{2} + b_i\right)$$

$$= (2k-1)!! 2^{k(l-1)} \sigma^{2k} \left( \frac{1}{2^m} \sum_{r=1}^{m} \binom{m}{r} \frac{\Gamma\left(k + \frac{r}{2}\right)}{\Gamma\left(\frac{r}{2}\right)} \right)^{l-1}. \tag{175}$$

For instance, for the variance ($k = 1$) the sum becomes:

$$\sum_{r_1=1}^{m} c_{r_1}^{1} \cdots \sum_{r_{l-1}=1}^{m} \binom{m}{r_{l-1}} \frac{1}{2^m \Gamma\left(\frac{r_{l-1}}{2}\right)} \prod_{i=1}^{l-1} \frac{1}{2} r_i \Gamma\left(\frac{1}{2} r_i\right) \tag{176}$$

$$= \sum_{r_1=1}^{m} \binom{m}{r_1} \frac{r_1}{2^{m+1}} \cdots \sum_{r_{l-1}=1}^{m} \binom{m}{r_{l-1}} \frac{r_{l-1}}{2^{m+1}}. \tag{177}$$

Now each sum can be solved independently:

$$\sum_{r_i=1}^{m} \binom{m}{r_i} \frac{r_i}{2^{m+1}} = \frac{1}{2^{m+1}} \left[ \sum_{r_i=1}^{m} \binom{m}{r_i} r_i \right] \tag{178}$$

$$= \frac{1}{2^{m+1}} \left[ m 2^{m-1} \right] \tag{179}$$

$$= \frac{m}{4}. \tag{180}$$

Therefore the variance is:

$$\mathbb{V}[Z] = \pi^{-\frac{1}{2}} 2^{\frac{l}{2}} (\sigma^2)^{-\frac{1}{2}} \left( \frac{1}{2^l \sigma^2} \right)^{-1-\frac{1}{2}} \Gamma\left(1 + \frac{1}{2}\right) \frac{m^{l-1}}{2^{2(l-1)}} \tag{181}$$

$$= \frac{1}{2} 2^{\frac{l}{2}} (\sigma^2)^{-\frac{1}{2}} \left( \frac{1}{2^{-\frac{3}{2}l} (\sigma^2)^{-\frac{3}{2}}} \right) \frac{m^{l-1}}{2^{2(l-1)}} \tag{182}$$

$$= \frac{1}{2} 2^l \sigma^2 \frac{m^{l-1}}{2^{2(l-1)}} \tag{183}$$

$$= \frac{\sigma^2 m^{l-1}}{2^{l-1}}. \tag{184}$$

Note how the variance of a ReLU net is significantly reduced if compared with the variance of a linear network of the same depth (compare with Eq. 78). Similarly, one can get the fourth moment:

$$\mathbb{E}[Z^4] = \frac{3(\sigma^2)^2 (m+5)^{l-1} m^{l-1}}{2^{2(l-1)}}. \tag{185}$$

Therefore the kurtosis is:

$$\kappa = 3 \left( \frac{m+5}{m} \right)^{l-1}. \tag{186}$$

Note how ReLU nets are more heavy-tailed than linear nets.

To calculate the asymptotic moments we need three technical Lemmas that express the quantities encountered in a better form. First we describe the coefficients of a factorized polynomial:

**Lemma C.3.** *Consider coefficients $a_1, \ldots, a_m \in \mathbb{R}$. Define the polynomial*

$$p(x) = \prod_{i=1}^{m} (x + a_i) = \sum_{i=1}^{m} \alpha_i x^i. \tag{187}$$

*Then it holds that $\alpha_m = 1$ and $\alpha_{m-1} = \sum_{i=1}^{m} a_i$.*

Next we use Lemma C.3 to write the ratio of Gamma functions as a polynomial:

**Lemma C.4.** *Fix $k \in \mathbb{N}$ and $x \in \mathbb{R}$. Then we can express the fraction of Gamma functions as follows:*

$$\frac{\Gamma(k + \frac{x}{2})}{\Gamma(\frac{x}{2})} = P_k(x) = \sum_{i=0}^{k} \alpha_i x^i, \tag{188}$$

*where $P_k$ is a $k$-th order polynomial with coefficients $\alpha_k = 2^{-k}$ and $\alpha_{k-1} = \frac{k^2 - k}{2^k}$.*

*Proof.* The leading coefficient can easily be obtained from multiplying together the terms $\frac{m}{2}$. From Lemma C.3 we conclude that

$$\alpha_{k-1} = \frac{1}{2^{k-1}} \sum_{i=1}^{k} (k-i) = \frac{1}{2^{k-1}} \left( k^2 - \frac{k(k+1)}{2} \right) = \frac{k^2 - k}{2^k}. \tag{189}$$

$\square$

Next we need to control the sums involving the factorials. Since we just expressed the ratio of Gamma functions as a polynomial, we essentially need to know how to control sums of the type

$$\frac{1}{2^m} \sum_{r=1}^{m} \binom{m}{r} r^k, \tag{190}$$

which amounts to controlling the moments of a binomial distribution with fault probability $p = \frac{1}{2}$. We do this as follows:

**Lemma C.5.** *Fix $k, m \in \mathbb{N}$. Then we can express the following sum as a polynomial $\forall k \in \mathbb{N}$:*

$$\frac{1}{2^m} \sum_{r=0}^{m} \binom{m}{r} r^k = \frac{m}{2^k} Q_{k-1}(m). \tag{191}$$

*where $Q_{k-1}$ is a $k-1$-th order polynomial. Moreover, writing $Q_l$ in monomial basis*

$$Q_l(m) = \sum_{i=0}^{l} \alpha_i m^i, \tag{192}$$

*it holds that $\alpha_l = 1$ and $\alpha_{l-1} = \frac{l(l+1)}{2} \ \forall l \in \mathbb{N}$.*

*Proof.* For a proof of the recursion, we refer to Boros and Moll (2004); Benyi (2005). Moreover the polynomials satisfy the recursion

$$Q_k(m) = 2m Q_{k-1}(m) - (m-1) Q_{k-1}(m-1). \tag{193}$$

Denote by $\alpha^{(k)}$ the coefficients of $Q_k$, so $\alpha_0^{(k)}, \ldots, \alpha_k^{(k)}$. Notice that the leading coefficient of $Q_k$ is thus $\alpha_k^{(k)}$ and for $Q_{k-1}$ it is $\alpha_{k-1}^{(k-1)}$. Using the recursion and performing a comparison of coefficients we see that

$$\alpha_k^{(k)} = 2\alpha_{k-1}^{(k-1)} - \alpha_{k-1}^{(k-1)} = \alpha_{k-1}^{(k-1)}. \tag{194}$$

Using the fact that for $k = 1$

$$\frac{1}{2^m} \sum_{r=0}^{m} \binom{m}{r} r = \frac{m}{2} = \frac{m}{2^1} Q_0(m), \tag{195}$$

we conclude that $\alpha_0^0 = 1$ and thus $\alpha_k = 1 \ \forall k \in \mathbb{N}$. For the second coefficient, namely $\alpha_{k-1}^{(k)}$ for $Q_k$ and $\alpha_{k-2}^{(k-1)}$ for $Q_{k-1}$, we will again use the recursion. Let us first again express the polynomials in monomial bases, i.e.

$$Q_k(m) = \sum_{i=0}^{k} \alpha_i^{(k)} m^i \ , \quad Q_{k-1}(m) = \sum_{i=0}^{k-1} \alpha_i^{(k-1)} m^i \tag{196}$$

Using the recursion we thus see that

$$\sum_{i=0}^{k} \alpha_i^{(k)} m^i = \sum_{i=0}^{k-1} 2\alpha_i^{(k-1)} m^{i+1} - \sum_{i=0}^{k-1} \alpha_i^{(k-1)} (m-1)^{i+1}. \tag{197}$$

We have to understand the terms involving $m^{k-1}$. Thus we need to expand $(m-1)^k$ which we can do with the help of Lemma C.3:

$$(m-1)^k = m^k - km^{k-1} + \ldots \tag{198}$$

We also need to expand the next polynomial as follows:

$$(m-1)^{k-1} = m^{k-1} + \ldots \tag{199}$$

Collecting all the coefficients, we end up with the following recursion for the second coefficient:

$$\begin{aligned} \alpha_{k-1}^{(k)} &= 2\alpha_{k-2}^{(k-1)} - \alpha_{k-2}^{(k-1)} + k\alpha_{k-1}^{(k-1)} \\ &= \alpha_{k-2}^{(k-1)} + k. \end{aligned} \tag{200}$$

Using the fact that

$$Q_1(m) = m + 1. \tag{201}$$

Thus $\alpha_0^{(1)} = 1$, we conclude that

$$\alpha_{k-1}^{(k)} = 1 + \sum_{i=2}^{k} i = \frac{k(k+1)}{2}. \tag{202}$$

$\square$

Finally, we need a result on exponential functions and their limit definition:

**Lemma C.6.** *Fix $c \in \mathbb{R}$ and $\gamma \in \mathbb{R}_+$. Then we have the following limit:*

$$\lim_{m \to \infty} \left( 1 + \frac{c}{m} + \mathcal{O}\left(\frac{1}{m^2}\right) \right)^{(\gamma m)} = e^{\gamma c}. \tag{203}$$

*Moreover, it holds that*

$$\lim_{m \to \infty} \left( 1 + \frac{c}{m} + \mathcal{O}\left(\frac{1}{m^2}\right) \right)^{m^\beta} = \begin{cases} \infty & \text{if } \beta > 1 \\ 1 & \text{if } \beta < 1 \end{cases} \tag{204}$$

*Proof.* This can be found in standard analysis books such as Rudin (1976). $\square$

### C.5 Proof of Theorem 5.1

We can now prove the convergence of the moments as follows.

**Theorem.** *Consider the distribution of the output $p\left(f_{ReLU}^{(L)}\right)$, as defined in Thm. 4.3. Denote $X \sim \mathcal{N}(0,1)$, $Y \sim \mathcal{LN}(-\frac{5}{4}\gamma, \frac{5}{4}\gamma)$ for $X \perp Y$ and the resulting normal log-normal mixture by $Z = XY$, for $\gamma > 0$. Let the depth grow as $L = c + \gamma m^\beta$ where $\beta \geq 0$ and $c \in \mathbb{N}$ fixed. Then it holds that for $k > 1$*

$$\mathbb{E}\left[\left(f_{ReLU}^{(L)}\right)^{2k}\right] \xrightarrow{m \to \infty} \begin{cases} \mathbb{E}[X^{2k}] = (2k-1)!! & \text{if } \beta < 1 \\ \mathbb{E}[Z^{2k}] = e^{\frac{5}{2}\gamma k(k-1)}(2k-1)!! & \text{if } \beta = 1 \\ \infty & \text{if } \beta > 1 \end{cases} \tag{205}$$

*where $(2k-1)!! = (2k-1)\ldots 3 \cdot 1$ denotes the double factorial (by symmetry, odd moments are zero). Moreover, for $\beta < 1$ it holds that*

$$p(f_{ReLU}^{(L)}) \xrightarrow{d} X \quad \text{for } m \to \infty \tag{206}$$

*Proof.* Recall that we arrived at

$$\mathbb{E}\left[\left(f_{\text{ReLU}}^{(l)}\right)^{2k}\right] = (2k-1)!!2^{k(l-1)}\sigma_w^{2kl}\left(\frac{1}{2^m}\sum_{r=1}^{m}\binom{m}{r}\frac{\Gamma\left(k+\frac{r}{2}\right)}{\Gamma\left(\frac{r}{2}\right)}\right)^{l-1}. \tag{207}$$

Using the NTK parametrization for ReLU, i.e. $\sigma_1^2 = 1$ and $\sigma_2^2 = \cdots = \sigma_l^2 = \frac{2}{m}$, this amounts to

$$\mathbb{E}\left[\left(f_{\text{ReLU}}^{(l)}\right)^{2k}\right] = (2k-1)!!\left(\frac{2^{2k}}{m^k 2^m}\sum_{r=1}^{m}\binom{m}{r}\frac{\Gamma\left(k+\frac{r}{2}\right)}{\Gamma\left(\frac{r}{2}\right)}\right)^{l-1}. \tag{208}$$

We thus essentially need to understand the term

$$M(m) = \frac{2^{2k}}{2^m m^k}\sum_{r=1}^{m}\binom{m}{r}\frac{\Gamma\left(k+\frac{r}{2}\right)}{\Gamma\left(\frac{r}{2}\right)} \tag{209}$$

We first use Lemma C.4 to expand the ratio $\frac{\Gamma\left(k+\frac{r}{2}\right)}{\Gamma\left(\frac{r}{2}\right)}$ as a polynomial. Denote the coefficients by $\beta_i$ for $i = 1, \ldots, k$ ($i \neq 0$ because the polynomial has no intercept). We then swap the two sums:

$$M(m) = \frac{2^{2k}}{2^m m^k}\sum_{r=1}^{m}\binom{m}{r}\sum_{i=1}^{k}r^i\beta_i = \frac{2^{2k}}{m^k}\sum_{i=1}^{k}\beta_i\frac{1}{2^m}\sum_{r=1}^{m}\binom{m}{r}r^i. \tag{210}$$

Now we can apply Lemma C.5 to expand the inner sum for each $i$, denoting the corresponding polynomials again by $Q_i$:

$$M(m) = \frac{2^{2k}}{m^k}\sum_{i=1}^{k}\beta_i\frac{m}{2^i}Q_{i-1}(m). \tag{211}$$

Notice that $mQ_{i-1}(m)$ is a polynomial of order $i$. For large $m$, the factor $\frac{1}{m^k}$ dominates all such polynomials except for the one with $i = k$. Thus in the large-width limit it holds

$$M(m) \xrightarrow{m\to\infty} 2^{2k}\beta_k\frac{1}{2^k} = 1. \tag{212}$$

where we used that the leading coefficient of $Q_{k-1}$ is 1. For fixed depth $l \in \mathbb{N}$ or depth growing as $l = m^\beta$ for $\beta < 1$, we can pull the limit $\lim_{m\to\infty}M(m)^{l-1}$ inside and conclude that

$$\mathbb{E}\left[\left(f_{\text{ReLU}}^{(l)}\right)^{2k}\right] \xrightarrow{m\to\infty} (2k-1)!!. \tag{213}$$

If $\beta > 1$, we obtain a divergence of the even moments for $k > 1$ to infinity as $m$ increases as the exponent grows faster than $m$. Note that for $k = 1$ however, we have that $\mathbb{E}\left[\left(f_{\text{ReLU}}^{(l)}\right)^2\right] = 1$ also in the $\beta > 1$ limit.

For depth growing as $l - 1 = \gamma m$ (so $\beta = 1$), we have to be a bit more careful since we need to compute the coefficient in front of $\frac{1}{m}$, similarly as in the linear case. We now need to collect all the polynomial terms in $M(m)$ giving rise to a $\frac{1}{m}$ factor. First recall that

$$M(m) = \frac{2^{2k}}{m^k}\sum_{i=1}^{k}\beta_i\frac{m}{2^i}Q_{i-1}(m). \tag{214}$$

The only coefficients contributing to $\frac{1}{m}$ are the second highest coefficient of $Q_{k-1}$ and the highest coefficient of $Q_{k-2}$. Using Lemma C.5 and Lemma C.4, we hence find that

$$\begin{aligned} M(m) &= 1 + 2^{2k}\left(\beta_k\frac{k(k-1)}{2}\frac{1}{2^k} + \beta_{k-1}\frac{1}{2^{k-1}}\right)\frac{1}{m} + \mathcal{O}\left(\frac{1}{m^2}\right) \\ &= 1 + 2^{2k}\left(\frac{1}{2^{2k+1}}(k-1)k + \frac{1}{2^{2k-1}}(k-1)k\right)\frac{1}{m} + \mathcal{O}\left(\frac{1}{m^2}\right) \\ &= 1 + \left(\frac{5((k-1)}{2}\right)\frac{1}{m} + \mathcal{O}\left(\frac{1}{m^2}\right). \end{aligned} \tag{215}$$

Applying C.6 concludes that

$$
\mathbb{E}\left[\left(f_{\text{ReLU}}^{(l)}\right)^{2k}\right] = (2k-1)!!\left(1 + \left(\frac{5k(k-1)}{2}\right)\frac{1}{m} + \mathcal{O}\left(\frac{1}{m^2}\right)\right)^{\gamma m} \tag{216}
$$
$$
\xrightarrow{m\to\infty} (2k-1)!!e^{\frac{5\gamma k(k-1)}{2}}.
$$

Finally, taking $X \sim \mathcal{N}(0,1)$, $Y \sim \mathcal{LN}(-\frac{5}{4}\gamma, \frac{5}{4}\gamma)$ and defining $Z = XY$, we can easily see that

$$
\mathbb{E}\left[Z^n\right] = \begin{cases} 0 & \text{n odd} \\ e^{\frac{5}{2}\gamma k(k-1)}(2k-1)!! & \text{n = 2k} \end{cases} \tag{217}
$$

$\square$

## D   Additonal results and lemmas

Here we list some of the moments arising from a Binomial distribution of the form $U \sim \text{Bin}(n, \frac{1}{2})$. We invite the reader to sanity-check our results in Lemma C.5 regarding the coefficients of $Q_k$.

**Lemma D.1.** *Consider the random variable $U \sim \text{Bin}(m, \frac{1}{2})$. We can calculate its first* 4 *moments as*

- $\mathbb{E}[U] = \sum_{i=0}^{m}\binom{m}{i}i = \frac{m}{2}Q_0(m) = \frac{m}{2}$
- $\mathbb{E}[U^2] = \sum_{i=0}^{m}\binom{m}{i}i^2 = \frac{m}{2^2}Q_1(m) = \frac{m}{2^2}(m+1)$
- $\mathbb{E}[U^3] = \sum_{i=0}^{m}\binom{m}{i}i^3 = \frac{m}{2^3}Q_2(m) = \frac{m}{2^3}(m^2+3m)$
- $\mathbb{E}[U^4] = \sum_{i=0}^{m}\binom{m}{i}i^4 = \frac{m}{2^4}Q_3(m) = \frac{m}{2^4}(m^3+6m^2+3m+4)$