# OpenReview forum: "Precise characterization of the prior predictive distribution of deep ReLU networks"
_NeurIPS.cc/2021/Conference — NeurIPS 2021 Spotlight_

### Official Review · Reviewer_E89g · 2021-07-03

**Rating:** 6
**Confidence:** 3

**Summary:**

Uses the Meijer-G function to analyse the properties of finite-width Bayesian neural networks.  Provides additional proofs of the infinite-width limit, in addition to results for the infinite depth limit.  Shows that finite networks are heavy tailed and that this increases with depth.

**Main Review:**

I really like the paper, the while the maths are a bit heavy and will be unfamiliar to most researchers, they do a fantastic job giving a clear high-level description of    what the Meijer-G function is and why it is useful to understand Bayesian neural networks.

My key issue surrounds references to past work.  The authors appear to be unfamiliar with a line of work that began with Aitchison (2019) "Why bigger is not always better:  on finite and infinite neural networks".  This work explicitly address the question of why infinite networks perform worse than finite networks (which is mentioned here in  the related work).  It concludes that finite networks differ from infinite networks principally in the existence of flexibility in the Gaussian process kernel, with deeper, narrower networks having more flexibility.  Flexibility/variability in the kernel naturally implies heavy-tailed-ness of the output distribution.  But more importantly, the view in terms of representations gives us much more insight into what the flexibility in finite nets is actually used for.  This work was continued by Aitchison el al. (2020) "Deep kernel processes", which uses these insights to develop a new family of flexible, deep kernel methods that are in some cases equivalent to deep neural networks, and give considerable  insights into representation learning in practically relevant deep networks.

There are a number of most likely contemporaneous works (potentially submitted to this conference).  While the authors may not have been aware of many of these (I realise that some may have appeared after the submission date to the conference), it is important to discuss the interrelationships between these works in the "Related Work" section.  These papers include:
* https://arxiv.org/abs/2104.11734
* https://arxiv.org/abs/2106.06529
* https://arxiv.org/abs/2106.00651
* https://arxiv.org/abs/2106.04013

Very happy to increase the score if these issues are resolved!

The "Predictive variance" section is a bit confusing.  It just seems to tell us how to set the variance of the weights of an infinite width in the network such that the output variance is a prescribed value e.g. tells us that a standard Gaussian prior gives extremely high output variance.  Unless I'm missing something, this has been obvious for quite a long time, and was the motivation behind the introduction of this the He normalisation.

**Time Spent Reviewing:**

3

---

> ### Author Response · Authors · 2021-08-10
> **Response**
>
> Thank you very much for your positive comments and taking the time to review our paper! We’re glad that you enjoyed reading our paper. We will address your concerns, mainly regarding related work, in the following.
>
> **1) “Why bigger is not always better: on finite and infinite neural networks”:**
>
> We are very glad that you pointed out the two works Aitchison (2019) "Why bigger is not always better: on finite and infinite neural networks" and Aitchison el al. (2020) "Deep kernel processes". We were unfortunately not aware of those papers. They are very relevant to our work as they explicitly study why finite neural networks can outperform their infinite counterpart, one of the main motivations for our work. In “Why bigger is not always better”, the author proves that the variance of the kernel is inversely proportional to the width of the net, and this causes a reduction in flexibility of the kernel that increasingly relies on the prior w.r.t. the data to make inference. The depth has the opposite effect, being proportional to the kernel variance. The connection you mention between flexibility/variability of the kernel with heavy-tailedness is very interesting and connects nicely with our result that deep networks become more heavy-tailed. In "Deep Kernel Processes", the authors try to fix the loss of flexibility issue by injecting stochasticity in the kernel through distributions over kernel matrices. This allows for more flexible kernels that have the NNGP kernel as special case. We would be very happy to use the additional page to discuss these works and their connections to ours in more depth!
>
> **2) “Other concurrent works”:**
>
> Thank you for pointing out these concurrent works. As you mention,
> most of these papers appeared shortly before or after we submitted our work but we are very happy to now use the chance to include them in the related works section. In the following, we will give a quick discussion of the papers you listed. We would provide more details in the final version of the paper.
>
> 2.a) **“Exact priors of finite neural networks”:** This concurrent work establishes a very similar result for the prior for finite width networks, also using the Meijer-G function. Our insights into the limiting behaviour of neural networks when both width and depth go to infinity, as well as the analytic characterization of heavy-tailedness however are unique to our work.
>
> 2.b) **“The limitations of large width in neural networks: a deep Gaussian process Perspective”:** This paper studies deep Gaussian processes priors and the authors prove that such priors converge to a single layer GP as width goes to infinity. They analyze the tails of the deep GP prior and find that increasing depth leads to non-Gaussian-like tails and increasing width leads to more Gaussian-like tails. This is in-line with our results for neural networks that also predict more non-Gaussian-like behaviour for increased depth. We however are able to give a stronger characterization of the behaviour for the special case of a neural network, linking the resulting distribution to a normal-lognormal mixture.
>
> 2.c) **“Asymptotics of representation learning in finite Bayesian neural networks”:**
> This paper studies finite-width corrections to the posterior for the class of linear neural
> networks and networks that allow for one layer with non-linearities. This is similar to our work, as we also aim to characterize the finite-width case, however we do so only for the prior by deriving its distribution directly instead of considering asymptotic series expansions.
>
> 2.d) **“The future is Log-Gaussian”:** This concurrent work establishes a similar result in the infinite depth and infinite width setting, showing that for Resnets, a lognormal behaviour emerges when the ratio of width and depth are kept constant. Our result is similar as we show that a normal-lognormal mixture emerges in the case of standard fully-connected networks. On the other hand, our main result is the characterization of the prior for finite width, through Meijer-G functions, which clearly distinguishes our work.
>
> **3) “Predictive variance”:**
>
> We agree, will improve this section to make it clearer. This section served as a self-contained guide on the consequences of the architectural choices on the shape of the resulting prior, using our theoretical results. Our goal was to highlight that using our analysis we can recover previous known results such as the He-initialization and extend them to the Generalized He-initialization, but in combination with our analysis on heavy-tailedness, provide a more complete picture of the prior. We absolutely agree that the predictive variance was known for quite some time and can be derived in a way simpler manner.: From a regularization perspective, a standard Gaussian prior implies that deeper layers units are, for sufficiently deep networks, significantly less regularized than the first layers. Whether this is concerning or not is left for future work. See also the answer for reviewer LLRC.

---

> > ### Comment · Area_Chair_zeEd · 2021-08-12
> > **Reviewer please reply and if warranted update review**
> >
> > Your AC

---

### Official Review · Reviewer_LLRC · 2021-07-16

**Rating:** 7
**Confidence:** 3

**Summary:**

Leveraging the Meijer-G function, the authors characterise the predictive priors of fully-connected (finite-width) feedforward induced by independent standard Gaussian weight priors. They show that they can recover existing results concerning infinite width and infinite depth networks, and are also able to tackle different limiting regimes.

**Main Review:**

The paper is very clearly written and the technical tools and derivations are introduced in a logical order. Although I was not intimately familiar with the Meijer-G function, I was able to follow the argument of the authors by reading the paper from start to finish without having to understand the details. I like, for example, that the authors started with the linear case, giving the readers the intuition needed to begin to understand the ReLU case.

Questions:
1. One future direction the authors mention is extending to other architectures, in particular, conv nets. What about other activation functions? Lemma C.1 especially leverages the piecewise linear nature of ReLU activations on line 616 of the Appendix. Can you also extend this to Leaky ReLU activations? What about Heaviside step activations?
2. Unless I missed it, I don't think there was any insight into the posterior that is induced. Is there anything you can see here?
3. This work exclusively applies to the Bayesian setup. What breaks in the NTK setup? I'm guessing the issue arises from the fact that the NTK is no longer close to constant for finite-width.
4. One part of the paper that confuses me is all of section 6. The authors derive the variance at the output of a finite-width network (equation (24)) by leveraging their previous results. As far as I can tell, this can be derived without the earlier parts of the paper, but perhaps I am missing something. The ReLU function is absolutely homogeneous, ReLU(|a|x) = |a|ReLU(x) for any real number a. Then, factor out the standard deviations from each of the weight matrices so that they are standard Gaussians. The result is a scalar product of all the standard deviations with a random variable with variance (1/2)^L, the same as the result obtained by the authors.
5. Is it possible to sample from this prior directly, without using a neural network?

**Time Spent Reviewing:**

2

---

> ### Author Response · Authors · 2021-08-10
> **Response**
>
> Thank you very much for your positive comments and taking the time to review our paper! We are very
> happy to hear that our exposition helped you to follow our theoretical results. We will address your questions one by one in the following.
>
> **1) “Other activations”:**
>
> Yes, our proof extends to any piece-wise linear activation, albeit that the results get significantly more involved for LeakyReLU for instance. Smooth activations such as Sigmoid or Tanh however require more theoretical tools as one can not easily reduce it back to the linear case.
>
> **2) “Insights into posterior”:**
>
> You are right, we did not derive any theoretical result for the posterior, other than the fact that we move away from Gaussian process priors for infinite depth infinite width, which implies that the posterior is also non-Gaussian. Our work however provides a first step towards this bigger goal by characterizing the prior and we hope that future work can build up on our results to derive insights into the posterior.
>
> **3) “NTK setting”:**
>
> In the NTK regime one would have to include the training dynamics (gradient descent). As you point out, in finite width we cannot profit from the fact that the kernel remains constant through-out optimization, making it difficult to derive insights into the function learnt gradient descent. It would be very interesting to understand on the other hand how the neural tangent kernel behaves when both width and depth go to infinity at the same time, in the same spirit as we considered in this work. We will leave exploring this question to future work.
>
> **4) “Predictive variance”:**
>
> We agree, and we will improve the section to make it clearer. Our goal was to highlight that using our analysis we can recover previous known results such as the He-initialization and extend them to the Generalized He-initialization. We absolutely agree that this can be proven in a clearly easier manner, without having to resort to our characterization of the prior. From a regularization perspective, a standard Gaussian prior implies that deeper layers units are, for sufficiently deep networks, significantly less regularized than the first layers. Whether this is concerning or not is left for future work.  Also, together with the section on heavy-tailedness, this section provides a self contained practical guide on the consequences of the architectural choice for fully connected nets. See also the answer for reviewer E89g.
>
> **5) “Sampling from the prior”:**
>
> Yes, one can solve the integral in the definition of the Meijer-G function numerically to sample from the prior. There are many libraries that implement such a procedure, we used the python library https://mpmath.org/. This works well for shallow networks but we found that for deep networks, sampling from the prior by propagating inputs through the network is often faster than using the algorithmic procedure to solve the integral.

---

### Official Review · Reviewer_55jz · 2021-07-16

**Rating:** 8
**Confidence:** 3

**Summary:**

The main result of the paper is a derivation of the exact characterization of priors in function space through Meijer G-functions. The results are obtained for Gaussian priors, ReLU and linear activation functions. The article also provides a solid analysis of the obtained characterization.

**Advantages**:
* It is the first accurate description of hidden units for standard feed-forward neural networks among the concurrent work mentioned in the paper.
* The results are in line with other works on the heavy-tailed nature of hidden units and the Gaussian process limit.
* There is a study on the role of width and depth demonstrating how wider networks lead to more Gaussian-like distributions.
* Authors also introduced generalized He priors that enable to tune the variance induced in the function space.


**Limitations**:
* Only linear and ReLU functions
* Only Gaussian priors
* Without the bias term


**Limitations And Societal Impact:**

The authors adequately addressed the limitations and potential negative societal impact of their work.

**Main Review:**


**Suggestions**:
* While there is a study on the depth limit, I think it is important to talk about the deep propagation papers and Edge of Chaos: Schoenholz et al. “Deep information propagation”. In addition,  Hayou et al “On the impact of the activation function on deep neural networks training” argue that smooth activation functions lead to better information propagation. Do you think it is possible to achieve similar results based on the obtained depth limit?
* I would like to see a discussion on the limitations of the setting and the Meijer-G functions approach such as if it is possible to add a bias term or to work with other activation functions (e.g. smooth function, see the previous paragraph).
* Lee et al. “Finite Versus Infinite Neural Networks: an Empirical Study” might be a relevant citation about the performance of finite-width networks over infinite-width counterparts
* Aitchison. (2020) “Why bigger is not always better: on finite and infinite neural networks.” might be added to the discussion

**Clarity**:

The paper is well-written and enjoyable to read despite the technicality of the results.

**Significance**:

I think the results are significant and will help to extend the line of research on function space description of Bayesian neural networks.

**Conclusion**:

The results are obtained for a simple model and seems hard to generalize to other settings or priors but they are very solid. I recommend publication.




**Minor**:

* “In-line” -> in line
* References: titles should be properly capitalised: Gaussian, NNGP, NTK, Meijer, Bayesian,… The conferences names should be unified. The reference of Novak et al is mentioned twice.

Appendix:
* “Based in” -> on
* “We state and proof”, “we proof”
* “For both distribution”
* “Well known” -> well-known
* “Is significantly reduce”


**Time Spent Reviewing:**

16

---

> ### Author Response · Authors · 2021-08-10
> **Response**
>
> Thank you very much for your positive comments and for spending such a significant time reviewing our
> paper! We are happy to hear that you enjoyed reading our paper despite its technical nature. We will address your suggestions and questions one by one in the following. In general, it is worth noting that with our approach we are able to study the predictive distribution when only a *single input datapoint* is propagated through the network, and not the stochastic process that arises from a sequence of inputs (as in NNGP). The kernel methods like NNGP are very relevant to our work, but additional tools are needed to study the finite-width counterpart. In this direction, see for instance Yaida (2019) “Non-Gaussian processes and neural networks at finite widths”.
>
> **1) “Deep Information Propagation and Smooth Activations”:**
>
> Thanks for pointing this out, those works are indeed relevant to ours! While the “Deep information propagation” paper uses a mean-field approach to establish depth scales at which information can still be propagated, we characterize the distribution exactly without any approximation. On the other hand, we have provided results only for a single input and we need more tools to understand how the correlation of two inputs, when propagated through the network, behaves as a function of depth. It would be very interesting to compare the behaviour of the correlation computed exactly, with the results obtained in “Deep information propagation” through the mean-field approach.
> So far, our results are limited to linear and Relu activation but can be extended to any piecewise linear activation, albeit that the resulting expressions are more involved. Understanding the infinite depth limit for smooth activations like tanh or sigmoid would be very interesting and is definitely something we will be working on in the future, but it will require some new theoretical tools as our proof relies on the piecewise nature of the activation. Knowledge of the resulting distribution at the output could indeed help with understanding which activation functions are more suitable for information propagation.
>
> **2) “Limitations”: Our analysis can easily include a bias term with Gaussian initialization.**
>
> Our proof can also be adapted for any piecewise linear activation function such as Leaky-ReLU, absolute value etc. We are currently also looking into placing a prior on the weights that is different from a Gaussian. Our approach relies on the fact that a Gaussian has an easy representation in terms of Meijer-G functions but in principle, this can potentially work for other distributions as well. This could thus potentially recover a broader range of results, including for instance “Stable behaviour of infinitely wide deep neural networks”. So far our results are also limited to fully-connected networks but, given the fact that the NNGP could also be derived for convolutional networks, we are hopeful that our approach can also scale to this setting.
>
> **3) “Finite Versus Infinite Neural Networks: an Empirical Study”:**
>
>  Thank you for pointing us to this work, we will include it in the discussion regarding finite vs infinite-width networks. Interestingly, as the authors implicitly point out in Section 3.3, single SGD-trained finite-width nets may underperform with respect to e.g. NNGP. This is because they are trained on mini-batches and the NNGP posterior leverages Bayesian methods (i.e. model ensembles). Although the paper doesn’t compare approximate inference like SG-MCMC to NNGP posterior, one can already see (Figure 1) that an ensemble of finite width nets almost closes this gap for FCN and outperforms infinite-width methods for CNN architectures. Indeed, this was also previously shown in works such as Garriga-Alonso et al, "Deep Convolutional Networks as shallow Gaussian Processes" and Novak at al "Bayesian Deep Convolutional Networks with Many Channels are Gaussian Processes".
>
> **4) “Why bigger is not always better: on finite and infinite neural networks”:**
>
>  Thank you for highlighting this work, we were not aware of it and it is very relevant to our work. We will add it to the related works sections and discuss it in more detail. Please see also the response to reviewer E89g for a discussion on this work.

---

### Decision · Program_Chairs · 2021-09-27

**Decision:**

Accept (Spotlight)

**Comment:**

This paper adds to the large and important body of work of characterising the prior in function space of a deep learning model implied by the architecture and the prior over the parameters. The paper focuses on Gaussian priors with relu and identity activation functions.

All the reviewers liked the paper. They found it clearly written and appreciated that the paper made an effort to convey the high level results despite the paper using quite heavy machinery (the Meijer-G function) to derive finite network width results. The discussion was substantial and led to a good consensus among reviewers and authors.

The paper adds substantial new insights into an important area of research in deep learning.

Accept. :-)